# Unsupervised Multi-Target Domain Adaptation: An Information Theoretic Approach

## Abstract

Unsupervised domain adaptation (**uDA**) models focus on pairwise adaptation settings where there is a single, labeled, source and a single target domain. However, in many real-world settings one seeks to adapt to multiple, but somewhat similar, target domains. Applying pairwise adaptation approaches to this setting may be suboptimal, as they fail to leverage shared information among multiple domains. In this work, we propose an information theoretic approach for domain adaptation in the novel context of multiple target domains with unlabeled instances and one source domain with labeled instances. Our model aims to find a shared latent space common to all domains, while simultaneously accounting for the remaining private, domain-specific factors. Disentanglement of shared and private information is accomplished using a unified information-theoretic approach, which also serves to establish a stronger link between the latent representations and the observed data. The resulting model, accompanied by an efficient optimization algorithm, allows simultaneous adaptation from a single source to multiple target domains. We test our approach on three challenging publicly-available datasets, showing that it outperforms several popular domain adaptation methods.

## 1 Introduction

In real-world data, the training and test instances often do not come from the same underlying distribution (Sun et al. (2016)). For example, in the task of object recognition/classification from image data, this may be due to the image noise, changes in the object view, etc., which induce different biases in the observed data sampled during the training and test stage. Consequently, assumptions made by traditional learning algorithms are often violated, resulting in degradation of the algorithms' performance during inference of test data. Domain Adaptation (**DA**) approaches (Fernando et al. (2013); Gong et al. (2012); Kodirov et al. (2015); Yoo et al. (2016)) aim to tackle this by transferring knowledge from a source domain (training data) to an unlabeled target domain (test data) to reduce the discrepancy between the source and target data distributions, typically by exploring domain-invariant data structures.

Existing **DA** methods tackle the adaptation problems in one of the two settings: (semi)supervised **DA** and unsupervised **DA** (Csurka (2017)). The former assume that in addition to the labeled data of the source domain, some labeled data from the target domain are also available for training/adapting the classifiers. In contrast, the latter does not require any labels from the target domain but rather explores the similarity in the data distributions of the two domains. In this work, we focus on the unsupervised DA (**uDA**) scenario, which is more challenging due to the lack of correspondences in source and target labels.

Most works on **uDA** today focus on a single-source-single-target-domain scenario. However, in many real-world applications, unlabeled data may come from different domains, thus, with different statistical properties but with common task-related content. For instance, we may have access to images of the same class of objects (e.g., cars) recorded by various types of cameras, and/or under different camera views and

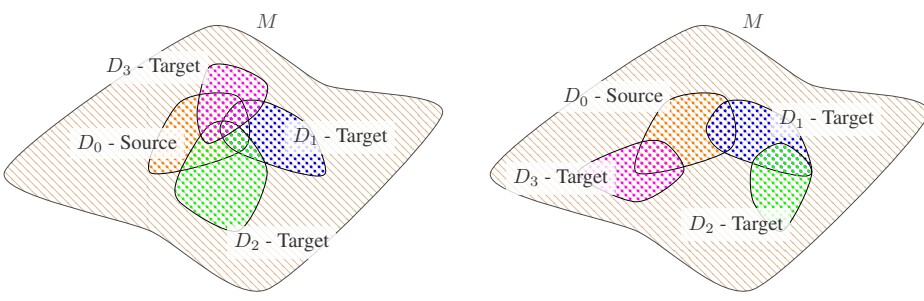

**(a)** Domains with common shared space.  **(b)** Domains with pairwise shared spaces.

**Figure 1:** Illustration of domains with common (a) and pairwise-shared spaces (b). We tackle the domain adaptation task when all domains share a common task/space, which is then leveraged to transfer knowledge across multiple target domains.

at different times, rendering multiple different domains (e.g., datasets). Likewise, facial expressions of emotions, such as joy and surprise, shown by different people and recorded under different views, result in multiple domains with varying data distributions. In most cases, these domains have similar *underlying* data distributions, which can be leveraged to build more effective and robust classifiers for tasks such as the object or emotion recognition across multiple datasets/domains. To this end, traditional **uDA** methods focus on the single-source-single-target **DA** scenario. However, in the presence of multiple domains, as typically encountered in real-world settings, this pair-wise adaptation approach may be suboptimal as it fails to leverage simultaneously the knowledge shared across multiple task-related domains. Recently, Zhao et al. (2017) showed that by having access to multiple source domains can facilitate better adaptation to a single target domain, when compared to the pair-wise **DA** approach. While this is intuitive due to the access to multiple *labelled* source domains, offering more adaptation flexibility for the target domain (i.e., by efficiently exploring the data labels across multiple source domains that are most related to the target domain), it comes at the expense of the data labelling in multiple source domains, which can be costly and time-consuming. In either case, a single source domain or readily available multiple source domains, to the best of our knowledge, a simultaneous adaptation to *multiple* and *unlabelled* target domains remains an unexplored **DA** scenario. However, this **DA** scenario is important as we usually have access to multiple unlabeled domains; yet, the adaptation process is also more challenging due to the lack of supervision in the target domains. Still, multi-target **DA** can have advantages over a single-target **DA** when: (i) there is direct knowledge sharing between the source and multiple target domains (fig. 1a), and (ii) the source and a target domain are related through another target domain (fig. 1b). While this seems intuitive, it is critical how the data from multiple *unlabelled* target domains are leveraged within the multi-target **DA** approach, in order to improve its performance over the single target **DA** approaches and naive fusion of multiple target domains.

To this end, we propose a Multi-Target DA-Information-Theoretic-Approach (**MTDA-ITA**) for single-source-multi-target **DA**. We exploit a single source domain and focus on multiple target domains to investigate the effects of multi-target **DA**; however, the proposed approach can easily be extended to multiple source domains. This approach leverages the data from multiple target domains to improve performance compared to individually learning from pair-wise source-target domains. Specifically, we simultaneously factorize the information from each available target domain and learn separate subspaces for modeling the shared (i.e., correlated across the domains) and private (i.e., independent between the domains) subspaces of the data (Salzmann et al. (2010)). To this end, we employ deep learning to derive an information theoretic approach where we jointly maximize the mutual information between the domain labels and private (domain-specific) features, while minimizing the mutual information between the the domain labels and the shared (domain-invariant) features. Consequently, the more robust feature representations are learned for each target domain by exploiting dependencies between multiple target domains. We show on benchmark datasets for **DA** that this approach leads to overall improved performance on each target domain, compared to independent **DA** for

each pair of source-target domains, or the naive combination of multiple target domains, and state-of-the-art models applicable to the target task.

## 2    THE PROPOSED METHOD

Without loss of generality, we consider a multi-class ($K$-class) classification problem as the running example. Furthermore, let $(\mathbf{X}, \mathbf{Y}, \mathbf{D}) = \{(\boldsymbol{x}_i, \boldsymbol{y}_i, \boldsymbol{d}_i)\}_{i=0}^{N}$ be a collection of $M$ domains (a labeled source domain, and $M - 1$ unlabeled target domains), where $\boldsymbol{x}_i$ denotes the $i$-th sample, and $\boldsymbol{y}_i = [y_i^1, y_i^2, ..., y_i^K]$ and $\boldsymbol{d}_i = [d_i^1, d_i^2, ..., d_i^M]$ are the $K$-D and $M$-D encoding of the class and domain labels for $\boldsymbol{x}_i$, respectively. Note that the class labels are only available for the source samples.

The latent space representation of the data point $\boldsymbol{x}$ is denoted as $\boldsymbol{z} = [\boldsymbol{z}_s, \boldsymbol{z}_p]$, where $\boldsymbol{z}_s$ and $\boldsymbol{z}_p$ are the (latent) shared and private features of the data point $\boldsymbol{x}$, respectively. By factorizing the joint distribution $p(\boldsymbol{x}, \boldsymbol{y}, \boldsymbol{d}, \boldsymbol{z}_s, \boldsymbol{z}_p)$ as

$$p(\boldsymbol{x}, \boldsymbol{y}, \boldsymbol{d}, \boldsymbol{z}_s, \boldsymbol{z}_p) = p(\boldsymbol{x})p(\boldsymbol{d})p(\boldsymbol{z}_s|\boldsymbol{x})p(\boldsymbol{z}_p|\boldsymbol{x})p(\boldsymbol{y}|\boldsymbol{z}_s), \tag{1}$$

we propose to maximize the following objective function:

$$\mathcal{L}(\theta_s, \theta_p, \theta_c; \boldsymbol{x}, \boldsymbol{y}, \boldsymbol{d}) = \lambda_r I(\boldsymbol{x}; \boldsymbol{z}) + \lambda_c I(\boldsymbol{y}; \boldsymbol{z}_s) + \lambda_d \big( I(\boldsymbol{d}; \boldsymbol{z}_p) - I(\boldsymbol{d}; \boldsymbol{z}_s) \big), \tag{2}$$

where $p(\boldsymbol{x})$ and $p(\boldsymbol{d})$ denote the underlying (true) data distribution and domain label distribution, respectively, $I(\boldsymbol{x}; \boldsymbol{y})$ denotes the Mutual Information between the random variables $\boldsymbol{x}$ and $\boldsymbol{y}$. $\lambda_r, \lambda_c$ and $\lambda_d$ denote the hyper-parameters controlling the weights of the objective terms. The proposed objective function (2) maximizes the three terms described below:

- $I(\boldsymbol{x}; \boldsymbol{z})$ : encourages the latent features (both shared and private) to preserve information about the data samples (that can be used to reconstruct $\boldsymbol{x}$ from $\boldsymbol{z}$).

- $I(\boldsymbol{y}; \boldsymbol{z}_s)$: enables to correctly predict the true class label of the samples out of their common shared features.

- $I(\boldsymbol{d}; \boldsymbol{z}_p) - I(\boldsymbol{d}; \boldsymbol{z}_s)$ : encourages the latent private features to preserve the information about the domain label and penalizes the latent shared features to be domain informative. This not only reduces the redundancy in the shared and private features, but also, penalizes the redundancy of different private spaces, while preserving the shared information.

An additional term could be used to minimize the mutual information between the shared ($\boldsymbol{z}_s$) and private ($\boldsymbol{z}_p$) features. However, computing the mutual information (even approximating it) is intractable due to the highly complex joint distribution $p(\boldsymbol{z}_s, \boldsymbol{z}_p)$. Since we want $\boldsymbol{z}_s$ and $\boldsymbol{z}_p$ features to encode different aspects of $\boldsymbol{x}$, we enforce such constraint by jointly maximizing the term: $I(\boldsymbol{d}; \boldsymbol{z}_p) - I(\boldsymbol{d}; \boldsymbol{z}_s)$.

### 2.1    OPTIMIZATION

The following lower bound for mutual information is derived using the non-negativity of KL-divergence (Barber & Agakov (2003)); i.e., $\Sigma_{\boldsymbol{x}} p(\boldsymbol{x}|\boldsymbol{z}) \ln \frac{p(\boldsymbol{x}|\boldsymbol{z})}{q(\boldsymbol{x}|\boldsymbol{z})} \geq 0$ gives:

$$I(\boldsymbol{x}; \boldsymbol{z}) \geq H(\boldsymbol{x}) + \mathbb{E}_{p(\boldsymbol{x}, \boldsymbol{z})}[\ln q(\boldsymbol{x}|\boldsymbol{z}; \phi)] \tag{3}$$

where $H(\boldsymbol{x})$ denotes the Shanon Entropy (Lin (1991)) of the random variable $\boldsymbol{x}$. $q(\boldsymbol{x}|\boldsymbol{z}; \phi)$ is any arbitrary distribution parameterized by $\phi$. We need a variational distribution $q(\boldsymbol{x}|\boldsymbol{z}; \phi)$ because the posterior distribution $p(\boldsymbol{x}|\boldsymbol{z}) = p(\boldsymbol{z}|\boldsymbol{x})p(\boldsymbol{x})/p(\boldsymbol{z})$ is intractable since the true data distribution $p(\boldsymbol{x})$ is assumed to be unknown. Similarly, we can derive lower bounds for $I(\boldsymbol{d}; \boldsymbol{z}_p) \geq H(\boldsymbol{d}) + \mathbb{E}_{p(\boldsymbol{d}, \boldsymbol{z}_p)}[\ln q(\boldsymbol{d}|\boldsymbol{z}_p; \psi)]$ and $I(\boldsymbol{d}; \boldsymbol{z}_s) \geq$

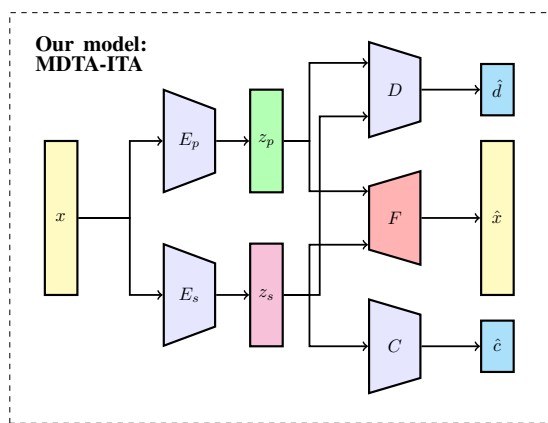

**Figure 2: MDTA-ITA**: The encoder $E_s(\boldsymbol{x})$ captures the feature representations ($z_s$) for a given input sample $\boldsymbol{x}$ that are shared among domains. $E_p(\boldsymbol{x})$ captures domain–specific private features ($z_p$) using the *shared* private encoder. The shared decoder $F(z_p, z_s)$ learns to reconstruct the input sample by using both the private and shared features. The domain classifier $D$ learns to correctly predict the domain labels of the actual samples from both their shared and private features while the classifier $C$ learns to correctly predict the class labels from the shared features.

$H(\boldsymbol{d}) + \mathbb{E}_{p(\boldsymbol{d},\boldsymbol{z}_s)}[\ln q(\boldsymbol{d}|\boldsymbol{z}_s; \psi)]$, where $q(\boldsymbol{d}|\boldsymbol{z}_p; \psi)$ is any arbitrary distribution parameterized by $\psi$.[1] We further drive lower bound for $I(\boldsymbol{y}; \boldsymbol{z}_s)$ as $I(\boldsymbol{y}; \boldsymbol{z}_s) \geq H(\boldsymbol{y}) + \mathbb{E}_{p(\boldsymbol{y},\boldsymbol{z}_s)}[\ln q(\boldsymbol{y}|\boldsymbol{z}_s; \theta_c)]$, where $q(\boldsymbol{y}|\boldsymbol{z}_s; \theta_c)$ is a variational distribution parameterized by $\theta_c$ approximating $p(\boldsymbol{y}|\boldsymbol{z}_s)$.

Let next $E_s(\boldsymbol{x}; \theta_s)$ be a function parameterized by $\theta_s$ that maps a sample $\boldsymbol{x}$ to its corresponding *shared* feature $\boldsymbol{z}_s$, and $E_p(\boldsymbol{x}; \theta_p)$ be an analogous function which maps $\boldsymbol{x}$ to $\boldsymbol{z}_p$, the feature that is *private* to each domain (fig. 2). We also define $F(\boldsymbol{z}_s, \boldsymbol{z}_p; \phi)$ as a decoding function mapping the concatenation of the latent features $\boldsymbol{z}_s$ and $\boldsymbol{z}_p$ to a sample reconstruction $\hat{\boldsymbol{x}}$, and $D(\boldsymbol{z}; \psi)$ as a decoding function mapping $\boldsymbol{z}_s$ and $\boldsymbol{z}_p$ to a $M$-dimensional vector: the predictions of the domain label $\hat{\boldsymbol{d}}$. Finally, $C(\boldsymbol{z}_s; \theta_c)$ is a task-specific function mapping $\boldsymbol{z}_s$ to a $K$-dimensional probability vector of the class label $\hat{\boldsymbol{y}}$.

We represent $p(\boldsymbol{d}), p(\boldsymbol{x}), p(\boldsymbol{y})$ as the empirical distribution of a finite training set (e.g. $p(\boldsymbol{d}) = \frac{1}{N} \sum_{i=1}^{N} \delta(\boldsymbol{d} - \boldsymbol{d}_i)$) as in the case of variational autoencoders (VAE) (Abbasnejad et al. (2017); Pu et al. (2017)), $p(\boldsymbol{z}_s|\boldsymbol{x}), p(\boldsymbol{z}_p|\boldsymbol{x})$ as deterministic functions of $\boldsymbol{x}$ as $p(\boldsymbol{z}_s|\boldsymbol{x}) = \delta(\boldsymbol{z}_s - E_s(\boldsymbol{x}; \theta_s))$ and $p(\boldsymbol{z}_p|\boldsymbol{x}) = \delta(\boldsymbol{z}_p - E_p(\boldsymbol{x}; \theta_p))$, and the variational distributions $q(\boldsymbol{y}|\boldsymbol{z}_s), q(\boldsymbol{x}|\boldsymbol{z})$ and $q(\boldsymbol{d}|\boldsymbol{z})$ as

$$q(\boldsymbol{y}|\boldsymbol{z}_s) = \text{SoftMax}(C(\boldsymbol{z}_s; \theta_c)), \quad q(\boldsymbol{d}|\boldsymbol{z}) = \text{SoftMax}(D(\boldsymbol{z}; \psi)), \quad q(\boldsymbol{x}|\boldsymbol{z}; \phi) \propto \exp(\|\boldsymbol{x} - F(\boldsymbol{z}; \phi)\|_1) \quad (4)$$

where $\text{Softmax}(\cdot)$ denotes the softmax or normalized exponential function (Bridle (1990)), and $\|.\|_1$ denotes the $L_1$ norm. Then, the optimization task can be posed as a minimax saddle point problem, where we use adversarial training to maximize (2) w.r.t. the parameters $(\theta_s, \theta_p, \theta_c)$, and to minimize (2) w.r.t. the parameters $(\phi, \psi)$, using Stochastic Gradient Descent (SGD).

**Optimizing the parameters $\phi$ of the decoder $F$**

$$\hat{\phi} = \arg\min_{\phi} \mathcal{L}_F = \frac{\lambda_r}{N} \sum_{i=1}^{N} \|\boldsymbol{x}_i - F(E_s(\boldsymbol{x}_i), E_p(\boldsymbol{x}_i))\|_1. \quad (5)$$

The decoder $F(\boldsymbol{z}_s, \boldsymbol{z}_p; \phi)$ is trained in such a way so as to minimize the difference between original input $\boldsymbol{x}$ and its decoding from corresponding shared and private features via the decoder $F$.

---

[1]Note that, for simplicity, we shared the parameters $\psi$ between the approximate posterior distributions $q(\boldsymbol{d}|\boldsymbol{z}_s, \psi)$ and $q(\boldsymbol{d}|\boldsymbol{z}_p; \psi)$.

**Optimizing the parameters $\psi$ of the domain classifier $D$**

$$\hat{\psi} = \arg\min_{\psi} \mathcal{L}_D = -\frac{\lambda_d}{N} \sum_{i=1}^{N} \boldsymbol{d}_i^\top \ln D\big(E_s(\boldsymbol{x}_i)\big) - \frac{\lambda_d}{N} \sum_{i=1}^{N} \boldsymbol{d}_i^\top \ln D\big(E_p(\boldsymbol{x}_i)\big). \tag{6}$$

$D(\boldsymbol{z}; \psi)$ can be considered as a classifier whose task is to distinguish between the shared/private features of the different domains. More precisely, the two terms in Eq. 6 encourage $D$ to correctly predict the domain labels from the shared and private features, respectively.

**Optimizing the parameters $\theta_c$ of the label classifier $C$**

$$\hat{\theta}_c = \arg\min_{\theta_c} \big\{ - H(\boldsymbol{y}) - \mathbb{E}_{p(\boldsymbol{y}, \boldsymbol{z}_s)}\big[ \ln q(\boldsymbol{y}|\boldsymbol{z}_s) \big] \big\}. \tag{7}$$

Since we have access to the source labels, $H(\boldsymbol{y})$ is a constant for source samples. we can approximate $H[\boldsymbol{y}]$ for the target samples using the output of the classifier $C$, leading to the following optimization problem:

$$\hat{\theta}_c = \arg\min_{\theta_c} \mathcal{L}_C = -\frac{1}{N} \sum_{i=1}^{N_s} \boldsymbol{y}_i^T \ln C\big(E_s(\boldsymbol{x}_i)\big) - \frac{\lambda_c}{N - N_s} \sum_{i=N_s+1}^{N} C\big(E_s(\boldsymbol{x}_i)\big)^\top \ln C\big(E_s(\boldsymbol{x}_i)\big)$$

$$+ \frac{\lambda_c}{N - N_s} \sum_{i=N_s+1}^{N} C\big(E_s(\boldsymbol{x}_i)\big)^\top \ln \bigg( \frac{1}{N - N_s} \sum_{i=N_s+1}^{N} C\big(E_s(\boldsymbol{x}_i)\big) \bigg), \tag{8}$$

where $N_s$ denotes the number of source samples. Intuitively, we enforce the classifier $C$ to correctly predict the class labels of the source samples by the first term in Eq. 8. We use the second term to minimize the entropy of $q(\boldsymbol{y}|\boldsymbol{z}_s)$ for the target samples; effectively, reducing the effects of "confusing" labels of target samples, as given by $p(\boldsymbol{y}|\boldsymbol{z}_s)$ that leads to decision boundaries occur far away from target data-dense regions in the feature space. The intuition behind the last term is that by minimizing only the entropy (second term), we may arrive at a degenerate solution where every target point $\boldsymbol{x}_t$ is assigned to the same class. Hence, the last term encourages the classifier $C$ to have balanced labeling for the target samples where it reaches its minimum, $\ln K$, when each class is selected with uniform probability.

**Optimizing the parameter $\theta_p$ of the private encoder $E_p$**

$$\hat{\theta}_p = \arg\min_{\theta_p} \mathcal{L}_P = \frac{\lambda_r}{N} \sum_{i=1}^{N} \|\boldsymbol{x}_i - F\big(E_s(\boldsymbol{x}_i), E_p(\boldsymbol{x}_i)\big)\|_1 - \frac{\lambda_d}{N} \sum_{i=1}^{N} \boldsymbol{d}_i^\top D\big(E_p(\boldsymbol{x}_i)\big). \tag{9}$$

The first term in Eq. 9 encourages the private encoder $E_p$ to preserve the recovery ability of the private features. The second term enforces distinct private features be produced for each domain by penalizing the representation redundancy in different private spaces. This, in turn, encourages moving this common information from multiple domains to their shared space.

**Optimizing the parameter $\theta_s$ of the shared encoder $E_s$**

$$\hat{\theta}_s = \arg\min_{\theta_s} \mathcal{L}_S = \frac{\lambda_r}{N} \sum_{i=1}^{N} \|\boldsymbol{x} - F\big(E_s(\boldsymbol{x}_i), E_p(\boldsymbol{x}_i)\big)\|_1 - \frac{\lambda_c}{N} \sum_{i=1}^{N_s} \boldsymbol{y}_i^T \ln C\big(E_s(\boldsymbol{x}_i)\big)$$

$$- \frac{\lambda_d}{N} \sum_{i=1}^{N} \boldsymbol{d}_i^\top \ln D\big(E_s(\boldsymbol{x}_i)\big) - \frac{\lambda_c}{N - N_s} \sum_{i=N_s+1}^{N} C\big(E_s(\boldsymbol{x}_i)\big)^\top \ln C\big(E_s(\boldsymbol{x}_i)\big)$$

$$+ \frac{\lambda_c}{N - N_s} \sum_{i=N_s+1}^{N} C\big(E_s(\boldsymbol{x}_i)\big)^\top \ln \bigg( \frac{1}{N - N_s} \sum_{i=N_s+1}^{N} C\big(E_s(\boldsymbol{x}_i)\big) \bigg). \tag{10}$$

The first term in Eq. 10 encourages the shared encoder $E_s$ to preserve the recovery ability of the shared features. The second term is the source domain classification loss penalty that encourages $E_s$ to produce discriminative features for the labeled source samples. The third term simulates the adversarial training by trying to fool the domain classifier $D$ when predicting the domain labels $d$, given the shared features $z_s$. The effect of this is two-fold: (i) the rendered shared features are more distinct from the corresponding private features, (ii) the shared features of different domains are encouraged to be similar to each other. The last two terms encourage $E_s$ to produce the shared features for target samples so that the classifier is confident on the unlabeled target data, driving the shared features away from the decision boundaries. To train our model, we alternate between updating the shared encoder $E_s$, the private encoder $E_p$, the decoder $F$, the classifier $C$, and the domain classifier $D$ using the SGD algorithm (see Algorithm 1 in Appendix E for more details).

## 3 RELATED WORK

There has been extensive prior work on domain adaptation (Csurka (2017)). Recent papers have focused on transferring deep neural network representations from a labeled source dataset to an unlabeled target domain, where the main strategy is to find a feature space such that the confusion between source and target distributions in that space is maximized ( Rebuffi et al. (2017); Benaim & Wolf (2017); Courty et al. (2017); Motiian et al. (2017); Saito et al. (2017); Zhang et al. (2017); Yan et al. (2017); Bousmalis et al. (2017)). For this, it is critical to first define a measure of divergence between source and target distributions. For instance, several methods have used the Maximum Mean Discrepancy (**MMD**) loss for this purpose (Bousmalis et al. (2017); Zellinger et al. (2017); Long et al. (2014)). **MMD** computes the norm of the difference between two domain means in the reproducing Kernel Hilbert Space (**RKHS**) induced by a pre-specified kernel. The Deep Adaptation Network (**DAN**) (Long et al. (2015)) applied **MMD** to layers embedded in a **RKHS**, effectively matching higher order statistics of the two distributions. The deep Correlation Alignment (**CORAL**) method (Sun & Saenko (2016)) attempts to match the mean and covariance of the two distributions. Deep Transfer Network (**DTN**) (Zhang et al. (2015)) achieved source/target distribution alignment via two types of network layers based on **MMD** distance: the shared feature extraction layer, which learns a subspace that matches the marginal distributions of the source and the target samples, and the discrimination layer, which matches the conditional distributions by classifier transduction.

Recently proposed unsupervised **DA** methods (Rebuffi et al. (2017); Benaim & Wolf (2017); Courty et al. (2017); Motiian et al. (2017); Saito et al. (2017); Zhang et al. (2017)) operate by training deep neural networks using adversarial training, which allows the learning of feature representations that are simultaneously discriminative of source labels, and indistinguishable between the source and target domain. For instance, Ganin & Lempitsky (2015) proposed a **DA** mechanism called Domain-Adversarial Training of Neural Networks (**DANN**), which enables the network to learn domain invariant representations in an adversarial way by adding a domain classifier and back-propagating inverse gradients. Adversarial Discriminative Domain Adaptation (**ADDA**) (Tzeng et al. (2017)) learns a discriminative feature subspace using the source labels, followed by a separate encoding of the target data to this subspace using an asymmetric mapping learned through a domain-adversarial loss. Liu et al. (2017) makes a shared-latent space assumption and proposes an unsupervised image-to-image translation (**UNIT**) framework based on Coupled GANs (Liu & Tuzel (2016)). Another example is the pixel-level domain adaptation models that perform the distribution alignment not in the feature space but directly in raw pixel space. **PixelDA** (Bousmalis et al. (2017)) uses adversarial approaches to adapt source-domain images as if drawn from the target domain while maintaining the original content.

While these approaches have shown success in **DA** tasks with single source-target domains, they are not designed to leverage information from multiple domains simultaneously. More recently, Zhao et al. (2017) introduced an adversarial framework called **MDAN** for multiple source single target domain adaptation where a domain classifier, induced by minimizing the H-divergence between multiple source and a target domain, is

used to align their feature distributions in a shared space. Instead, in our approach we focus on multi-target **DA** where we perform adaptation of multiple *unlabelled* target domains. Although both our model and **MDAN** use the similar notion of the domain classifier to minimize the domain mismatch in shared space, the domain classifier induced by our information-theoretic (IT) loss also acts to separate domains in the private space (see Eqs. 6 & 9 for more details), improving the essential reconstruction ability, similar to (Bousmalis et al. (2016)). We provided how our model is related to IT representation learning approaches, and multiple domain transfer networks in Appendices A and B respectively. In Appendix C, we also clearly contrasted our model with **DSN** model which also uses the notion of auto-encoders to explicitly separate the feature representations private to each source/target domain from those that are shared between the domains.

## 4 EXPERIMENTAL RESULTS

We compare the proposed method with state-of-the-art methods on standard benchmark datasets: a digit classification task that includes 4 datasets: **MNIST** (LeCun et al. (1998)), **MNIST-M** (Ganin et al. (2016)), **SVHN** (Netzer et al. (2011)), **USPS** (Tzeng et al. (2017)), **Multi-PIE** expression recognition dataset[2], and **PACS** multi-domain image recognition benchmark (Li et al. (2017)), a new dataset designed for the cross-domain recognition problems (the details for this experiment is available in Appendix G). Fig. 3 illustrates image samples from different datasets and domains. We evaluate the performance of all methods with classification accuracy metric. We repeated each experiment 5 times and report the average and the standard deviation of the accuracy.

We used ADAM (Kingma & Ba (2015)) for training; the learning rate was set to $0.0002$ and momentum parameters to $0.5$ and $0.999$. We used batches of size 16 from each domain, and the input images were mean-centered/rescaled to $[-1, 1]$. The hyper-parameters are empirically set as $\lambda_r = 1.0, \lambda_c = 0.01, \lambda_d = 0.20$. For the network architecture, our private/shared encoders consisted of three convolutional layers as the front-end and four basic residual blocks as the back-end. The decoder consisted of four basic residual blocks as the front-end and four transposed convolutional layers as the back-end. The discriminator and the classifier consisted of stacks of convolutional layers. We used ReLU for nonlinearity. Tanh function is used as the activation function of the last layer in the decoder $F$ for scaling the output pixels to $[-1, 1]$. The details of the networks are given in Appendix D.

The quantitative evaluation involves a comparison of the performance of our model to previous work and to **Source Only** and **1-NN** baselines that do not use any domain adaptation. For **Source Only** baseline, we train our model only on the unaltered source training data and evaluate on the target test data. We compare the proposed method **MTDA-ITA** with several related methods designed for pair-wise source-target adaptation: **CORAL** (Sun & Saenko (2016)), **DANN** (Ganin & Lempitsky (2015)), **ADDA** (Tzeng et al. (2017)), **DTN** (Zhang et al. (2015)), **UNIT** (Liu et al. (2017)), **PixelDA** (Bousmalis et al. (2017)), and **DSN** (Bousmalis et al. (2016)). We reported the results of two following baselines: (i) one is to combine all the target domains into a single one and train it using **MTDA-ITA**, which we denote as (**c-MTDA-ITA**). (ii) the other one is to train multiple **MTDA-ITA** separately, where each one corresponds to a source-target pair which we denote as (**s-MTDA-ITA**). For completeness, we reported the results of the competing methods by combining all the target domains into a single one (denoted by **c-DTN**, **c-ADDA**, and **c-DSN**) as well. We also extend **DSN** to multiple domains by (i) having one private encoder for all domains denoted by (**1p-DSN**), (ii) adding multiple private encoders to it denoted by (**mp-DSN**) and contrast them with our model.

### 4.1 DIGITS DATASETS

We combine four popular digits datasets (**MNIST**, **MNIST-M**, **SVHN**, and **USPS**) to build the multi-target domain dataset. All images were uniformly rescaled to $32 \times 32$. We take each of **MNIST-M**,

---

[2]http://www.cs.cmu.edu/afs/cs/project/PIE/MultiPie/Multi-Pie/Home.html

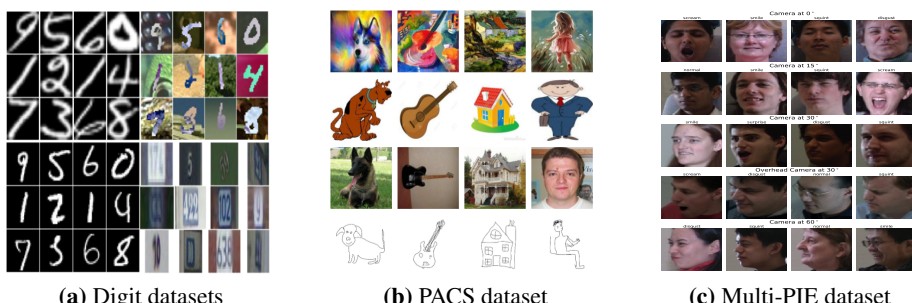

| **(a)** Digit datasets | **(b)** PACS dataset | **(c)** Multi-PIE dataset |

**Figure 3:** Exemplary images from different datasets. a) Digits datasets, b) **PACS** datatset (first row: Art-painting, second row: Cartoon, Third row: Photo, last row: Sketch), c) **Multi-PIE** dataset (each row corresponds to a different camera angle and each subject depicts an expression(**normal**, **smile**, **surprise**, **squint**, **disgust**, **scream**) at every camera position).

**SVHN**, **USPS**, and **MNIST** as source domain in turn, and the rest as targets. We use all labeled source images and all unlabeled target images, following the standard evaluation protocol for unsupervised domain adaptation (Ganin et al. (2016); Long et al. (2016)). We show the accuracy of different methods in Table 1. Additional results are available in Appendix F. The results show that first of all **cMTDA-ITA** has worse performance than **sMTDA-ITA** and **MTDA-ITA**. We have similar observations for **ADDA**, **DTN**, and **DSN** that demonstrates a naive combination of different target datasets can sometimes even decrease the performance of the competing methods. Furthermore, **MTDA-ITA** outperforms the state-of-the-art methods in most of domain transformations. The higher performance of **MTDA-ITA** compared to other methods is mainly attributed to the joint adaptation of related domains where each domain could benefit of other related domains. Furthermore, from the results obtained, we see that it is beneficial to use information coming from unlabeled target data (see Eq. 8 for updating the classifier $C$) during the learning process, compared to when no data from target domain is used (See the ablation study section for more information). Indeed, using our scheme, we find a representation space in which embeds the knowledge from the target domain into the learned classifier. By contrast, the competing methods do not provide a principled way of sharing information across all domains, leading to overall lower performance. The results also verify the superiority of **MTDA-ITA** over both **mp-DSN**, and **1p-DSN**. This can be due to (i) having multiple private encoders increase the number of parameters that may lead to **mp-DSN** overfitting, (ii) superiority of the **MTDA-ITA**'s domain adversarial loss over the **DSN**'s **MMD** loss to separate the shared and private features, (iii) utilization of the unlabeled target data to regularize the classifier in **MTDA-ITA**.

## 4.2 MULTI-PIE DATASET

The **Multi-PIE** dataset includes face images of 337 individuals captured from different expressions, views, and illumination conditions (fig. 3(c)). For this experiment, we use 5 different camera views (positions) $C05$, $C08$, $C09$, $C13$, and $C14$ as different domains (Fig. 3(c)) and the face expressions (**normal, smile, surprise, squint, disgust, scream**) as labels. Each domain contains 27120 images of size $64 \times 64 \times 3$. We used each view as the source domain, in turn, and the rest as targets. We expect the face inclination angle to reflect the complexity of transfer learning. Tab. 2 shows the classification accuracy for $C13$ and $C14$ as source domain (the results for views $C05, C08$ and $C09$ as source domain are available in Appendix F). As can be seen, **MTDA-ITA** achieves the best performances as well as the best scores in most settings that verifies the effectiveness of **MTDA-ITA** for multi-target domain adaptation. Clearly, with the increasing camera angle, the image structure changes up to a certain extent (the views become heterogeneous). However, our method produces better results even under such very challenging conditions.

| method | S → M | S → MM | S → U | M → S | M → MM | M → U | Ave. ranking |
|---|---|---|---|---|---|---|---|
| **Source Only** | 62.10 ± 0.60 | 40.43 ± 0.70 | 39.90 ± 0.60 | 30.29 ± 0.59 | 55.98 ± 0.48 | 78.30 ± 0.38 | 14.00 |
| **1-NN** | 35.86 | 18.21 | 29.31 | 28.01 | 12.58 | 41.22 | 15.00 |
| **CORAL** (Sun & Saenko (2016)) | 63.10 ± 0.61 | 54.37 ± 0.53 | 50.15 ± 0.63 | 33.40 ± 0.74 | 57.70 ± 0.69 | 81.05 ± 0.80 | 11.33 |
| **DANN** (Ganin et al. (2016)) | 73.80 ± 0.49 | 61.05 ± 0.80 | 62.54 ± 0.91 | 35.50 ± 0.65 | 77.40 ± 0.73 | 81.60 ± 0.60 | 8.75 |
| **ADDA** (Tzeng et al. (2017)) | 77.68 ± 0.92 | 64.23 ± 0.70 | 64.10 ± 0.79 | 30.04 ± 0.98 | 91.47 ± 1.0 | 90.51 ± 0.80 | 6.43 |
| **c-ADDA** | 80.10 ± 0.69 | 56.80 ± 0.79 | 64.80 ± 0.88 | 27.50 ± 0.86 | 83.30 ± 0.90 | 84.10 ± 0.98 | 8.95 |
| **DTN** (Zhang et al. (2015)) | 81.40 ± 0.42 | 63.70 ± 0.39 | 60.12 ± 0.52 | 40.40 ± 0.50 | 85.70 ± 0.39 | 85.80 ± 0.46 | 6.04 |
| **c-DTN** | 82.10 ± 0.62 | 59.30 ± 0.59 | 56.87 ± 0.65 | 38.32 ± 0.50 | 80.90 ± 0.80 | 79.31 ± 0.78 | 7.96 |
| **PixelDA** (Bousmalis et al. (2017)) | – | – | – | – | 98.10* | 94.10* | – |
| **UNIT** (Liu et al. (2017)) | 90.6* | – | – | – | – | 92.90 | – |
| **DSN** (Bousmalis et al. (2016)) | 82.70 ± 0.37 | 64.80 ± 0.40 | 65.30 ± 0.28 | 49.30 ± 0.30 | 83.20 ± 0.30 | 91.65 ± 0.40 | 2.85 |
| **c-DSN** | 83.10 ± 0.20 | 60.56 ± 0.36 | 60.35 ± 0.59 | 46.80 ± 0.45 | 80.49 ± 0.40 | 88.21 ± 0.38 | 4.84 |
| **1p-DSN** | 81.00 ± 0.47 | 58.22 ± 0.68 | 58.06 ± 0.48 | 45.11 ± 0.33 | 77.33 ± 0.52 | 85.16 ± 0.63 | 4.90 |
| **mp-DSN** | 83.40 ± 0.30 | 61.00 ± 0.50 | 58.10 ± 0.64 | 47.35 ± 0.40 | 79.30 ± 0.59 | 86.45 ± 0.71 | 5.33 |
| **s-MTDA-ITA** | 82.90 ± 0.13 | 63.10 ± 0.28 | 63.54 ± 0.30 | 49.60 ± 0.25 | 82.42 ± 0.19 | 89.21 ± 0.28 | 2.88 |
| **c-MTDA-ITA** | 79.20 ± 0.28 | 59.90 ± 0.30 | 63.70 ± 0.26 | 45.30 ± 0.30 | 77.12 ± 0.22 | 87.47 ± 0.25 | 4.25 |
| **MTDA-ITA** | **84.60** ± 0.24 | **65.30** ± 0.15 | **70.03** ± 0.20 | **52.01** ± 0.21 | 85.50 ± 0.18 | **94.20** ± 0.20 | **1.16** |

**Table 1:** Classification results on digit datasets. M: **MNIST**; MM: **MNIST-M**, S: **SVHN**, U: **USPS**. The best is shown in red. c-X: combining all target domains into a single one and train it using X. **s-MTDA-ITA**: training multiple **MTDA-ITA** where each one correspond to a source-target pair. **1p-DSN**: extended **DSN** with single private encoder. **mp-DSN**: extended **DSN** with multiple private encoder. Last column shows the average rank of each method over all adaptation pairs. ***UNIT** trains with the extended **SVHN** ($> 500$K images vs ours 72K). ***PixelDA** uses ($\approx 1,000$) of labeled target domain data as a validation set for tuning the hyper-parameters.

| method | C13 → C05 | C13 → C08 | C13 → C09 | C13 → C14 | C14 → C05 | C14 → C08 | C14 → C09 | C14 → C13 | Ave. ranking |
|---|---|---|---|---|---|---|---|---|---|
| **Source Only** | 50.79 ± 0.33 | 45.90 ± 0.50 | 40.04 ± 0.40 | 59.68 ± 0.29 | 60.03 ± 0.55 | 36.80 ± 0.61 | 40.11 ± 0.50 | 60.57 ± 0.36 | 16.08 |
| **1-NN** | 33.21 | 37.01 | 34.45 | 48.79 | 47.44 | 28.24 | 30.86 | 44.86 | 17.00 |
| **CORAL** | 54.89 ± 0.52 | 48.90 ± 0.48 | 40.30 ± 0.53 | 68.90 ± 0.35 | 59.98 ± 0.45 | 40.63 ± 0.55 | 40.80 ± 0.53 | 65.11 ± 0.45 | 11.95 |
| **DANN** | 57.86 ± 0.41 | 50.30 ± 0.43 | 45.30 ± 0.50 | 70.68 ± 0.35 | 57.20 ± 0.45 | 40.22 ± 0.55 | 40.77 ± 0.45 | 70.50 ± 0.55 | 9.92 |
| **ADDA** | 64.83 ± 0.69 | 63.20 ± 0.45 | 55.48 ± 0.65 | 74.25 ± 0.55 | 73.62 ± 0.75 | 43.56 ± 0.95 | 38.68 ± 0.95 | 72.84 ± 0.75 | 9.33 |
| **c-ADDA** | 59.20 ± 0.25 | 30.70 ± 0.63 | 53.20 ± 0.40 | 68.33 ± 0.35 | 65.88 ± 0.38 | 30.60 ± 0.61 | 45.34 ± 0.48 | 64.30 ± 0.40 | 11.50 |
| **DTN** | 63.78 ± 0.29 | 60.45 ± 0.35 | 60.55 ± 0.35 | 72.60 ± 0.25 | 70.67 ± 0.30 | 41.55 ± 0.65 | 41.45 ± 0.45 | 70.67 ± 0.45 | 8.75 |
| **c-DTN** | 57.53 ± 0.42 | 55.24 ± 0.45 | 57.14 ± 0.39 | 65.16 ± 0.35 | 63.80 ± 0.42 | 38.97 ± 0.71 | 39.80 ± 0.65 | 62.10 ± 0.45 | 10.92 |
| **PixelDA** | 45.68 ± 0.52 | 44.95 ± 0.42 | 44.45 ± 0.55 | **90.50** ± 0.25 | 46.28 ± 0.60 | 45.89 ± 0.61 | 44.45 ± 0.51 | 69.15 ± 0.45 | 9.95 |
| **UNIT** | 44.14 ± 0.10 | 44.47 ± 0.11 | 44.21 ± 0.12 | 44.47 ± 0.11 | 43.03 ± 0.1 | 44.44 ± 0.15 | 44.47 ± 0.15 | 44.47 ± 0.05 | 11.07 |
| **DSN** | 64.15 ± 0.30 | 57.70 ± 0.38 | 49.15 ± 0.45 | 80.75 ± 0.27 | 82.20 ± 0.28 | 38.75 ± 0.53 | 45.00 ± 0.25 | 80.50 ± 0.35 | 5.15 |
| **c-DSN** | 57.34 ± 0.45 | 31.63 ± 0.60 | 51.17 ± 0.40 | 74.52 ± 0.37 | 82.01 ± 0.35 | 34.25 ± 0.58 | 42.63 ± 0.55 | 79.42 ± 0.35 | 8.20 |
| **1p-DSN** | 55.84 ± 0.50 | 30.03 ± 0.50 | 49.06 ± 0.38 | 72.11 ± 0.50 | 81.22 ± 0.45 | 33.33 ± 0.58 | 42.03 ± 0.24 | 78.78 ± 0.57 | 8.63 |
| **mp-DSN** | 55.20 ± 0.46 | 30.40 ± 0.50 | 47.80 ± 0.35 | 75.30 ± 0.25 | 80.75 ± 0.20 | 30.20 ± 0.55 | 43.00 ± 0.35 | 79.02 ± 0.40 | 8.88 |
| **s-MTDA-ITA** | 70.10 ± 0.27 | 58.90 ± 0.25 | 58.10 ± 0.27 | 80.12 ± 0.15 | 82.05 ± 0.18 | 45.90 ± 0.30 | 52.67 ± 0.30 | 81.60 ± 0.24 | 3.65 |
| **c-MTDA-ITA** | 60.34 ± 0.17 | 55.67 ± 0.21 | 57.10 ± 0.23 | 73.50 ± 0.20 | 76.80 ± 0.10 | 43.10 ± 0.12 | 48.10 ± 0.14 | 80.90 ± 0.11 | 5.01 |
| **MTDA-ITA** | **78.40** ± 0.2 | **66.70** ± 0.17 | **70.30** ± 0.14 | 85.49 ± 0.11 | **87.20** ± 0.10 | **61.40** ± 0.14 | **60.05** ± 0.13 | **86.70** ± 0.10 | **1.20** |

**Table 2:** Classification results on Multi-PIE dataset. Last column shows the average rank of each method over all adaptation pairs. The best is shown in red.

## 4.3 Ablation Studies

We performed an ablation study on the proposed model measuring impact of various terms on the model's performance. To this end, we conducted additional experiments for the digit datasets with different components ablation, i.e., training without the reconstruction loss (denoted as **MTDA-woR**) by setting $\lambda_r = 0$, training without the classifier entropy loss (denoted as **MTDA-woE**) by setting $\lambda_c = 0$, training without the multi-domain separation loss (denoted as **MTDA-woD**) by setting $\lambda_d = 0$.

As can be seen from fig. 4, disabling each of the above components leads to degraded performance. More precisely, the average drop by disabling the classifier entropy loss is $\approx 3.5\%$. Similarly, by disabling the reconstruction loss and the multi-domain separation loss, we have $\approx 4.5\%$ and $\approx 22\%$ average drop in performance, respectively. Clearly, by disabling the multi-domain separation loss, the accuracy drops

significantly due to the severe data distribution mismatch between different domains. The figure also demonstrates that leveraging the unlabeled data from multiple target domains during training enhances the generalization ability of the model that leads to higher performance. In addition, the performance drop caused by removing the reconstruction loss , i.e., without the private encoder/decoder, indicates (i) the benefit of modeling the latent features as the combination of shared and private features, (ii) the ability of the model's domain adversarial loss to effectively learn those features.

In order to examine the effect of the private features on the model's classification performance, we took the **MTDA-ITA** and trained it without the private encoder (denoted as **MTDA-woP**). As fig. 4 shows, without the private features, the model performed consistently worse ($\approx 2\%$ average drop in performance) in all scenarios. This demonstrates explicitly modeling what is unique to each domain can improve the model's ability to extract domain–invariant features. In summary, this ablation study showed that the individual components bring complimentary information to achieve the best classification results.

### 4.4 Feature Visualization

We use t-SNE (Maaten & Hinton (2008)) on Digit dataset to visualize shared and private feature representations from different domains. Fig. 6 shows shared and private features from source (**SVHN**) and target domains before (a),(b) and after adaptation (c),(d). **MTDA-ITA** significantly reduces the domain mismatch for the shared features (circle markers in fig. 6d, strong mixing of domain labels in this cluster, fig. 6c) and increases it for the private features (triangle markers, pure and well-separated domain clusters in fig. 6c). This is partially due to the proposed multi-domain separation loss through the use of the domain classifier $D$, which penalizes the domain mismatch for the shared features and rewards the mismatch for the private features. Moreover, as supported by the quantitative results in tab. 1, joint adaptation of related domains and the classifier, accomplished through the model, leads to superior class separability, compared to original features. This is depicted in fig. 6d, where the points in the shared space (large cluster) are grouped into class-specific subgroups (color indicates class label), while they are mixed in private spaces (smaller clusters). This is in contrast to fig. 6b, where original features show no class-specificity.

We also show the learned shared and private features for the models **MTDA-woE**, **MTDA-woP**, **MTDA-woR**, and **MTDA-woD**, in figs. 6e to 6l. Note that since the private encoder $E_p$ is disabled for **MTDA-woR**, and **MTDA-woP**, no private features are depicted in figs. 6g to 6j. The class label separation in the shared space for **MTDA-woE**, **MTDA-woP**, and **MTDA-woR**, figs. 6f, 6h and 6j, is still evident but not as strong as in the full model, fig. 6d, corroborating the small loss in classification accuracy observed in fig. 4a. On the other hand, **MTDA-woD** has significant mixing of class labels in the shared space, fig. 6l, more so than **MTDA-woE**, **MTDA-woR**, and **MTDA-woP**, implying worse classification prediction in fig. 4a due to the severe mismatch between different domains. Since our model uses one private encoder for all target domains, we also contrasted the visualization of **DSN** model with one private encoder **1p-DSN** in Appendix H.

## 5 Conclusion

This paper presented an information theoretic end-to-end approach to **uDA** in the context of a single source and multiple target domains that share a common task or properties. The proposed method learns feature representations invariant under multiple domain shifts and simultaneously discriminative for the learning task. This is accomplished by explicitly separating representations private to each domain and shared between source and target domains using a novel discrimination strategy. Our use of a single private domain encoder results in a highly scalable model, easily optimized using established back-propagation approaches. Results on three benchmark datasets for image classification show superiority of the proposed method compared to the state-of-the-art methods for unsupervised domain adaptation of visual domain categories.

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

## A    CONNECTION TO INFORMATION THEORETIC REPRESENTATION LEARNING

The idea of using information theoretic (IT) objectives for representation learning was originally introduced in (Tishby & Zaslavsky (2015)). Since their approach for optimizing the IT objective functions relied on the iterative Blahut Arimoto algorithm (Tishby & Zaslavsky (2015)), it is not feasible to apply to deep neural network (DNN) frameworks. Similar to our approach, there have been some recent works (Mohamed & Rezende (2015); Chalk et al. (2016); Alemi et al. (2018b; 2016; 2018a)) to approximate the MI by applying variational bounds on **MI**, though not in the context of domain adaptation.

 Mohamed & Rezende (2015) utilized the variational bounds on **MI**, and apply it to DNNs in the context of reinforcement learning. Chalk et al. (2016) and Alemi et al. (2016), developed the same variational lower bound In the context of Information Bottleneck (IB) principle (Tishby & Zaslavsky (2015)), where the former applied it to sparse coding problems, and used the kernel trick to achieve nonlinear mappings, whereas the latter applied it to DNNs to handle large datasets thanks to the SGD algorithm. Achille & Soatto (2018) proposed a variational bound on the **MI** in the context of IB, from the perspective of variational dropout and demonstrated its utility in learning disentangled representations for variational autoencoders.

The main difference between our method and the above methods is that these methods throw away the information in the data not related to the task by minimizing the mutual information between the data points and the latent representations that may lead to ignoring the individual characteristics (private features) of the datasets in a multiple dataset regime, whereas our method explicitly models what is unique to each domain (dataset) that improves the model's ability to extract domain–invariant features.

In the unsupervised representation learning literature, our work is also related to the VAE-based models (Bowman et al. (2015)). However, we propose to tackle the task using our IT approach using deterministic mappings instead of the traditional evidence lower bound (**ELBO**) optimization with stochastic mappings. In contrast to the unsupervised representation learning approaches, our setting also allows us to further improve the latent representation using the labeled data in the source domain while leveraging the sharing of dependencies across different target domains.

## B    CONNECTION TO MULTIPLE DOMAIN TRANSFER NETWORKS

Recent studies have shown remarkable success in multiple domain transfer (MDT) (Choi et al. (2017); Anoosheh et al. (2017); Kameoka et al. (2018); Hao et al. (2018)) though not in the context of the image classification, rather in the context of image generation. Choi et al. (2017) proposed **StarGAN**, a generative adversarial network capable of learning mappings among multiple domains in the contest of image to image translation framework. The goal of **StarGAN** is to train a single generator $G$ though this requires passing in a vector along with each input to the generator specifying the output domain desired, that learns mappings among multiple domains. To achieve this, $G$ is trained to translate an input image $x$ into an output image $x'$ conditioned on the target domain label $d$, $G(x, d) \rightarrow x'$. Similar to our domain classifier module $D$, they introduce an auxiliary classifier that allows a single discriminator to control multiple domains.

 Anoosheh et al. (2017) introduced **ComboGAN**, which decouples the domains and networks from each other. Similar to our encoder/decoder modules, **ComboGAN**'s generator networks contain encoder/decoders assigning each encoder and decoder to a domain. They combine the encoders and decoders of the trained model like building blocks, taking as input any domain and outputting any other. For example during

inference, to transform an image $x$ from an arbitrary domain $\mathbf{X}$ to $x'$ from domain $\mathbf{X}'$, they simply perform $x' = G_{\mathbf{X}',\mathbf{X}}(x) = Decoder_{X'}(Encoder_X(x))$. The result of $Encoder_X(x)$ can even be cached when translating to other domains as not to repeat computation.

The main differences between the MDT methods and ours is that, unlike our method which does domain alignment in feature space, MDT methods adapt representations not in feature space but rather in raw pixel space; translating samples from one domain to the "style" of a other domains. This works well for limited domain shifts where the domains are similar in pixel-space, but can be too limiting for settings with larger domain shifts that results in poor performance in significant structural change of the samples in different domains.

## C  Connection to Domain Separation Networks

The method closest to our work is Domain Separation Networks (**DSN**) (Bousmalis et al. (2016)), which use the notion of auto-encoders to explicitly separate the feature representations private to each source/target domain from those that are shared between the domains. Although extending **DSN** to multiple domains might seem trivial, **DSN** requires an autoencoder per domain, making the model impractical in the case of more than a couple of domains.

The overall loss of **DSN** consists of a reconstruction loss for each domain modeled by a shared decoder, a similarity loss such as **MMD**, which encourages domain invariance modeled by a shared encoder, and a dissimilarity loss modeled by two private encoders: one for the source domain and one for the target domain. While one could attempt to generalize **DSN** to multiple target domains by having individual per-target domain private encoders, doing so would prove problematic when the number of target domains is large — each private encoder would require a large "private" dataset to learn the private parameters. Precisely, for multiple ($M$) target domains, we could train a **DSN** model with one shared encoder, $M + 1$ private encoder (one for each domain), and one shared decoder. This leads to $M + 3$ models to train that implies the number of models increases linearly with the number of domains, as does the required training time. Second, **DSN** uses an orthogonality constraint among the shared and the private representations which may not be strong enough to remove redundancy and enforce disentangling among different private spaces. Precisely, **DSN** defines the loss via a soft subspace orthogonality constraint between the private and shared representation of each domain. However, it does not enforce the private representation of different domains to be different that may result in redundancy of different private spaces.

In addition, **DSN** enforces separation of spaces using the notion of Euclidean orthogonality, e.g., $\|z_s - z_p\|^2$. In case of multiple target domains, this would result in learning of all pairs of private spaces independently. To address those deficiencies, we first explicitly couple different private encoders into a single private encoder model, $E_{\theta_p}$ of fig. 2 , which allows us to generalize to an arbitrary number of target domains. To assure that the information among the private and shared spaces is not shared (i.e., "orthogonal"), we define an information-theoretic criteria enforced by a domain classifier, $D_\psi$ of fig. 2, which aims to segment the private space into clusters that correspond to individual target domains. By using $D_\psi$ within the adversarial framework, **MTDA-ITA** learns simultaneously the shared and private features from different domains (see fig. 6). We showed in Sec. 4 that our model performs better than the trivial extension of DSNs to the multi-domain case.

## D  Network Architecture

The network architecture used for the experiments is given in tab. 3. We use the following abbreviation for ease of presentation: N=Neurons, K=Kernel size, S=Stride size, D=Number of Domains, C=number of Classes. The transposed convolutional layer is denoted by DCONV. The residual basic block is denoted as RESBLK.

| Layer | Encoders (shared, private) |
|---|---|
| 1 | CONV-(N16,K7,S1), ReLU |
| 2 | CONV-(N32,K3,S2), ReLU |
| 3 | CONV-(N64,K3,S2), ReLU |
| 4 | RESBLK-(N64,K3,S1) |
| 5 | RESBLK-(N64,K3,S1) |
| 6 | RESBLK-(N64,K3,S1) |
| 7 | RESBLK-(N64,K3,S1) |

| Layer | Decoder |
|---|---|
| 1 | RESBLK-(N64,K3,S1) |
| 2 | RESBLK-(N64,K3,S1) |
| 3 | RESBLK-(N64,K3,S1) |
| 4 | RESBLK-(N64,K3,S1) |
| 5 | DCONV-(N32,K3,S2), ReLU |
| 6 | DCONV-(N16,K3,S2), ReLU |
| 7 | DCONV-(N1,K1,S1), Tanh |

| Layer | Discriminator |
|---|---|
| 1 | CONV-(N4,K3,S1), ReLU |
| 2 | CONV-(N8,K3,S1), ReLU |
| 3 | CONV-(N16,K3,S1), ReLU |
| 4 | CONV-(N32,K3,S1), ReLU |
| 5 | CONV-(N1,K3,S1), ReLU |
| 6 | DENSE-(ND), Softmax |

| Layer | Classifier |
|---|---|
| 1 | CONV-(N4,K3,S1), ReLU |
| 2 | CONV-(N8,K3,S1), ReLU |
| 3 | CONV-(N16,K3,S1), ReLU |
| 4 | CONV-(N32,K3,S1), ReLU |
| 5 | CONV-(N1,K3,S1), ReLU |
| 6 | DENSE-(NC), Softmax |

**Table 3:** Network architecture for the experiments.

## E  PROPOSED MODEL'S ALGORITHM

The detailed optimization process of the proposed model is shown in Algorithm 1.

## F  ADDITIONAL EXPERIMENTS FOR DIGIT AND MULTI-PIE DATASETS

The additional experiments for Digit dataset where we set **MNIST-M** and **USPS** as source domain is available in tab. 4. The additional experiments for Multi-PIE dataset where we set $C05$, $C08$ and $C09$ as source domain is available in tabs. 5 & 6.

## G  PACS DATASET

This dataset contains 9991 images ($227 \times 227 \times 3$ dimension) across 7 categories ('dog', 'elephant', 'giraffe','guitar', 'house', 'horse' and 'person') and 4 domains of different stylistic depictions ('Photo', 'Art painting', 'Cartoon' and 'Sketch'). The very diverse depiction styles provide a significant gap between

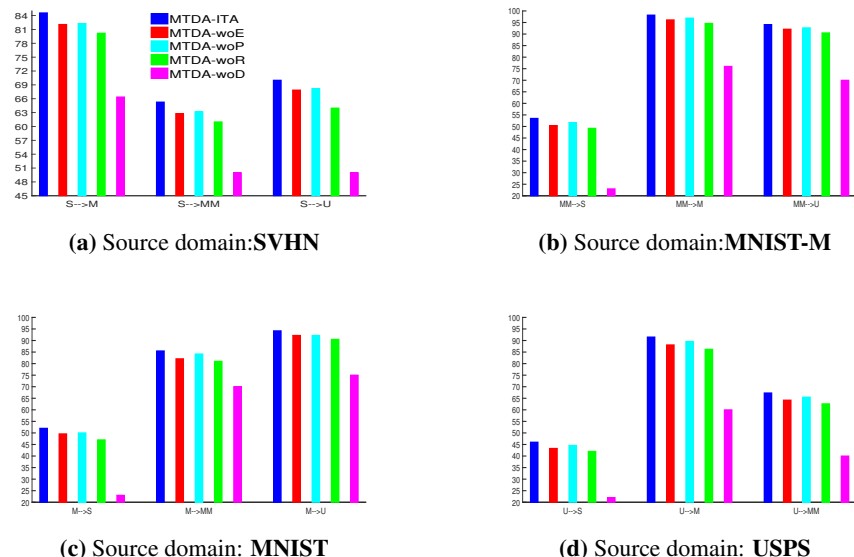

**Figure 4:** Ablation of **MTDA-ITA** on Digit dataset. We show that each component of our method, Reconstruction loss, Classifier entropy loss with separating shared/private features, contributes to the overall performance.

| method | MM → S | MM → M | MM → U | U → S | U → M | U → MM |
|---|---|---|---|---|---|---|
| **Source Only** | $40.00 \pm 0.61$ | $84.46 \pm 0.29$ | $80.43 \pm 0.50$ | $23.41 \pm 0.52$ | $50.64 \pm 0.37$ | $41.45 \pm 0.38$ |
| **1-NN** | 21.45 | 82.13 | 36.90 | 15.34 | 38.45 | 18.54 |
| **CORAL** (Sun & Saenko (2016)) | $40.20 \pm 0.60$ | $84.90 \pm 0.70$ | $87.54 \pm 0.44$ | $38.90 \pm 0.96$ | $85.01 \pm 0.61$ | $60.45 \pm 0.70$ |
| **DANN** (Ganin et al. (2016)) | $51.80 \pm 0.91$ | $61.05 \pm 0.71$ | $85.34 \pm 0.64$ | $35.50 \pm 0.84$ | $77.40 \pm 0.64$ | $61.60 \pm 0.64$ |
| **ADDA** (Tzeng et al. (2017)) | $40.64 \pm 0.86$ | $92.82 \pm 0.48$ | $80.70 \pm 0.48$ | $41.23 \pm 0.78$ | $90.10 \pm 0.58$ | $56.21 \pm 0.79$ |
| **c-ADDA** | $35.43 \pm 0.94$ | $88.47 \pm 0.61$ | $74.19 \pm 0.58$ | $39.36 \pm 0.99$ | $84.67 \pm 0.94$ | $52.54 \pm 0.88$ |
| **DTN** (Zhang et al. (2015)) | $48.80 \pm 0.66$ | $88.80 \pm 0.38$ | $90.68 \pm 0.35$ | $42.43 \pm 0.61$ | $89.04 \pm 0.36$ | $55.78 \pm 0.40$ |
| **c-DTN** | $44.21 \pm 0.61$ | $83.60 \pm 0.54$ | $84.98 \pm 0.41$ | $39.75 \pm 0.64$ | $85.04 \pm 0.45$ | $48.86 \pm 0.54$ |
| **UNIT** (Liu et al. (2017)) | – | – | – | | 90.60 | – |
| **DSN** (Bousmalis et al. (2016)) | $51.50 \pm 0.64$ | $90.20 \pm 0.31$ | $89.95 \pm 0.29$ | $\textcolor{red}{48.20} \pm 0.59$ | $\textcolor{red}{91.40} \pm 0.30$ | $60.45 \pm 0.35$ |
| **c-DSN** | $47.10 \pm 0.50$ | $84.60 \pm 0.40$ | $84.80 \pm 0.39$ | $40.50 \pm 0.61$ | $86.05 \pm 0.46$ | $56.25 \pm 0.50$ |
| **1p-DSN** | $45.00 \pm 0.60$ | $81.96 \pm 0.60$ | $83.03 \pm 0.49$ | $39.30 \pm 0.51$ | $84.55 \pm 0.56$ | $55.03 \pm 0.60$ |
| **mp-DSN** | $47.15 \pm 0.64$ | $85.51 \pm 0.54$ | $83.24 \pm 0.24$ | $38.30 \pm 0.74$ | $87.40 \pm 0.35$ | $55.47 \pm 0.44$ |
| **s-MTDA-ITA** | $50.55 \pm 0.18$ | $94.82 \pm 0.21$ | $89.05 \pm 0.28$ | $40.13 \pm 0.30$ | $87.10 \pm 0.25$ | $61.01 \pm 0.24$ |
| **c-MTDA-ITA** | $47.32 \pm 0.19$ | $90.20 \pm 0.30$ | $90.01 \pm 0.24$ | $41.10 \pm 0.35$ | $85.35 \pm 0.28$ | $60.31 \pm 0.34$ |
| **MTDA-ITA** | $\textcolor{red}{53.50} \pm 0.22$ | $\textcolor{red}{98.20} \pm 0.10$ | $\textcolor{red}{94.10} \pm 0.11$ | $46.00 \pm 0.48$ | $\textcolor{red}{91.50} \pm 0.23$ | $\textcolor{red}{67.30} \pm 0.15$ |

**Table 4:** Classification results on digit datasets. M: **MNIST**; MM: **MNIST-M**, S: **SVHN**, U: **USPS**. The best is shown in red. c-X: combining all target domains into a single one and train it using X. **s-MTDA-ITA**: training multiple **MTDA-ITA** where each one correspond to a source-target pair. **mp-DSN**: extended **DSN** with multiple private encoder. *UNIT trains with the extended **SVHN** ($> 500K$ images vs ours 72K). *PixelDA uses ($\approx 1,000$) of labeled target domain data as a validation set for tuning the hyper-parameters.

domains, coupled with the small number of data samples, making it extremely challenging for domain adaptation. Consequently, the dataset was originally used for multi-source to single target domain adaptation (Li et al. (2017)). Instead, we tackle a significantly more challenging problem of single-source to multiple target adaptation. Tab. 7 shows the classification accuracy of various methods. **MTDA-ITA** consistently

| method | C05 → C08 | C05 → C09 | C05 → C13 | C05 → C14 |
|---|---|---|---|---|
| **Source Only** | $31.56 \pm 0.40$ | $40.67 \pm 0.36$ | $39.89 \pm 0.22$ | $54.70 \pm 0.25$ |
| **1-NN** | 27.28 | 31.22 | 33.66 | 47.04 |
| **CORAL** (Sun & Saenko (2016)) | $36.55 \pm 0.66$ | $38.60 \pm 0.67$ | $40.60 \pm 0.58$ | $55.29 \pm 0.47$ |
| **DANN** (citeganin2016domain) | $40.30 \pm 0.60$ | $41.20 \pm 0.65$ | $40.12 \pm 0.60$ | $58.90 \pm 0.38$ |
| **ADDA** (Tzeng et al. (2017)) | $33.21 \pm 0.81$ | $30.86 \pm 0.90$ | $52.44 \pm 0.80$ | $70.18 \pm 0.60$ |
| **c-ADDA** | $46.88 \pm 0.65$ | $36.38 \pm 0.88$ | $39.14 \pm 0.85$ | $65.41 \pm 0.69$ |
| **DTN** (Zhang et al. (2015)) | $38.50 \pm 0.51$ | $30.56 \pm 0.46$ | $55.78 \pm 0.36$ | $68.90 \pm 0.31$ |
| **c-DTN** | $41.70 \pm 0.42$ | $31.10 \pm 0.48$ | $50.19 \pm 0.45$ | $60.34 \pm 0.35$ |
| **PixelDA** (Bousmalis et al. (2017)) | $44.93 \pm 0.42$ | $44.75 \pm 0.45$ | $45.18 \pm 0.45$ | $46.88 \pm 0.49$ |
| **UNIT** (Liu et al. (2017)) | $44.47 \pm 0.21$ | $44.47 \pm 0.21$ | $44.47 \pm 0.20$ | $44.51 \pm 0.28$ |
| **DSN** (Bousmalis et al. (2016)) | $45.12 \pm 0.46$ | $44.35 \pm 0.49$ | $48.12 \pm 0.53$ | $75.00 \pm 0.39$ |
| **c-DSN** | $42.52 \pm 0.48$ | $38.54 \pm 0.64$ | $34.15 \pm 0.64$ | $69.45 \pm 0.55$ |
| **1p-DSN** | $41.64 \pm 0.58$ | $37.84 \pm 0.63$ | $34.65 \pm 0.44$ | $68.75 \pm 0.85$ |
| **mp-DSN** | $41.30 \pm 0.28$ | $35.14 \pm 0.35$ | $34.40 \pm 0.35$ | $65.70 \pm 0.27$ |
| **s-MTDA-ITA** | $44.40 \pm 0.23$ | $44.60 \pm 0.25$ | $47.65 \pm 0.27$ | $80.20 \pm 0.13$ |
| **c-MTDA-ITA** | $40.49 \pm 0.25$ | $40.70 \pm 0.25$ | $42.80 \pm 0.25$ | $71.60 \pm 0.10$ |
| **MTDA-ITA** | $\mathbf{49.01} \pm 0.20$ | $\mathbf{48.20} \pm 0.27$ | $\mathbf{53.13} \pm 0.22$ | $\mathbf{84.29} \pm 0.10$ |

**Table 5:** Classification results on Multi-PIE dataset. The best is shown in red.

---

**Algorithm 1** MDTA-ITA Algorithm

---

**Require:** $\{\mathbf{X}, \mathbf{Y}, \mathbf{D}\}$:M domain datasets.
$\qquad \lambda_r, \lambda_c, \lambda_d$: Model hyper-parameters.
**Ensure:** $\theta_s, \theta_p, \theta_c, \phi, \psi$: Model parameters.
 1: Initialize $\theta_s, \theta_p, \theta_c, \phi, \psi$;
 2: **repeat**
 3:  Sample a mini-batch from each of source/target domain datasets.
 4:  Update $\{\theta_s\}$ by minimizing $\mathcal{L}_s$ in Eq.(10) through the gradient descent: $\theta_s = \theta_s - \eta \frac{\partial \mathcal{L}_s}{\partial \theta_s}$.
 5:  Update $\{\theta_p\}$ by minimizing $\mathcal{L}_p$ in Eq.(9) through the gradient descent:$\theta_p = \theta_p - \eta \frac{\partial \mathcal{L}_p}{\partial \theta_p}$.
 6:  Update $\{\theta_c\}$ by minimizing $\mathcal{L}_c$ in Eq.(8) through the gradient descent:$\theta_c = \theta_c - \eta \frac{\partial \mathcal{L}_c}{\partial \theta_c}$.
 7:  Update $\{\phi\}$ by minimizing $\mathcal{L}_\phi$ in Eq.(6) through the gradient descent:$\phi = \phi - \eta \frac{\partial \mathcal{L}_\phi}{\partial \phi}$.
 8:  Update $\{\psi\}$ by minimizing $\mathcal{L}_\psi$ in Eq.(5) through the gradient descent:$\psi = \psi - \eta \frac{\partial \mathcal{L}_s}{\partial \psi}$.
 9: **until** Convergence;
10: return $\{\theta_s, \theta_p, \theta_c, \phi, \psi\}$.

---

achieves the best performance for all transfer tasks. Evaluations were obtained by training all models (**ADDA**, **DSN**, and ours) from scratch on the **PACS** dataset. Note that the overall performance figures are low due to the extreme difficulty of the transfer task, induced by large differences among domains.

## H    ANALYSIS OF SHARED/PRIVATE SPACE EMBEDDING

In the experiments conducted, we showed that our approach is able to achieve better performance than the competing methods including the extended **DSN** with one private encoder (**1p-DSN**) which is the most similar method to ours.

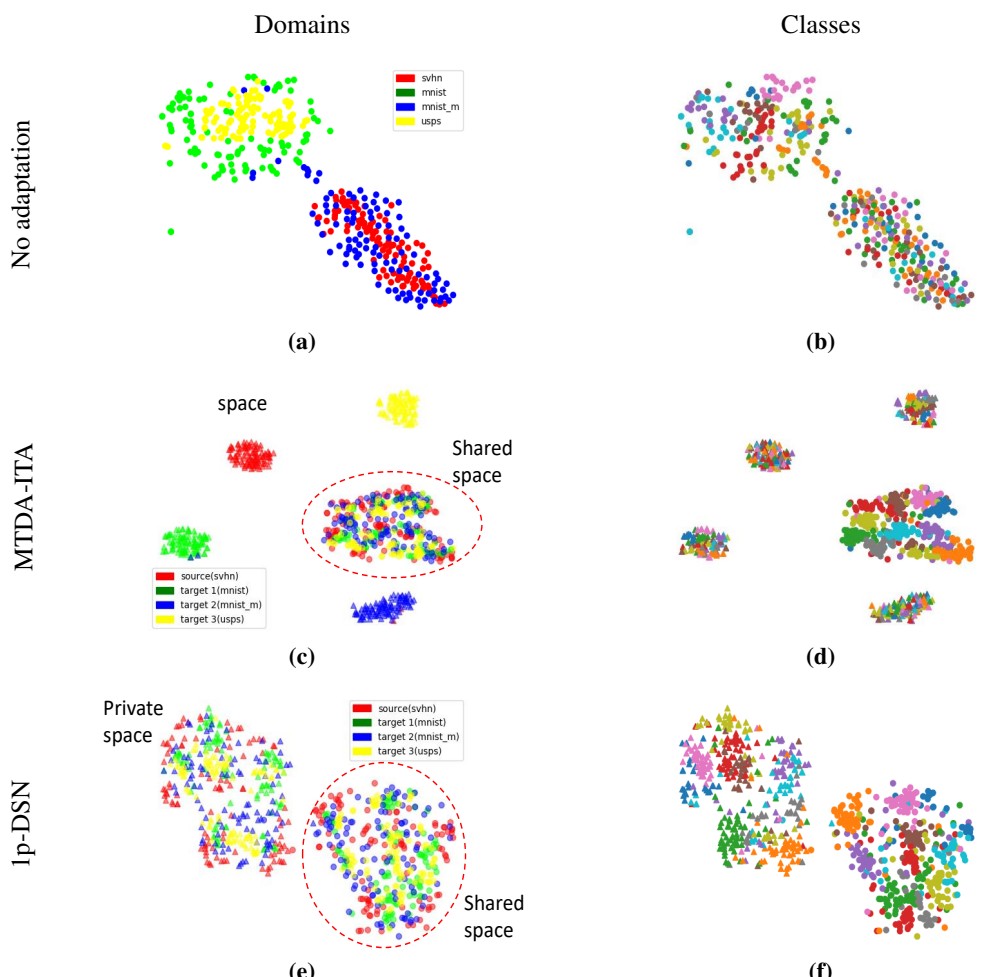

**Figure 5:** Feature visualization for embedding of digit datasets using t-SNE algorithm. The first and the second columns show the domains and classes, respectively, with color indicating domain and class membership. (a),(b) Original features. (c),(d) learned features for **MTDA-ITA** (triangle marker: private features, circle marker: shared features). Large clusters in the right column represent points from the shared space, while the smaller ones are from the private spaces. (e),(f) learned features for **1p-DSN**.

Indeed, fig. 5 depicts the embedding of the **MTDA-ITA** learned private/shared features, those of **1p-DSN** and the original features from different domains for Digit datasets (**SVHN** is the source).

Notice that both **MTDA-ITA** and **1p-DSN** reduces the domain mismatch for the shared features (circle markers in fig. 5) and separate the shared features from private features. On the other hand, **MTDA-ITA** increases the domain separation for the private features (triangle markers, pure and well-separated domain clusters in fig. 5c) while **1p-DSN** is unable to enforce the private representation of different domains to be different (fig. 5e) that may result in redundancy of different private spaces. This is partially due to the proposed multi-domain separation loss through the use of the domain classifier $D$, which penalizes the domain mismatch for the shared features and rewards the mismatch for the private features, something the **1p-DSN** fails to account for. Moreover, as supported by the quantitative results in tab. 1, the class label separation in

| method | C08 → C05 | C08 → C09 | C08 → C13 | C08 → C14 | C09 → C05 | C09 → C08 | C09 → C13 | C09 → C14 |
|---|---|---|---|---|---|---|---|---|
| **Source Only** | 33.70 ± 0.33 | 50.10 ± 0.28 | 50.80 ± 0.32 | 40.13 ± 0.26 | 33.32 ± 0.44 | 48.24 ± 0.32 | 49.24 ± 0.30 | 36.19 ± 0.27 |
| **1-NN** | 28.75 | 35.39 | 39.79 | 32.13 | 26.82 | 35.30 | 34.26 | 28.41 |
| **CORAL** (Sun & Saenko (2016)) | 35.89 ± 0.44 | 55.79 ± 0.50 | 60.00 ± 0.29 | 40.67 ± 0.48 | 35.89 ± 0.40 | 51.56 ± 0.43 | 50.45 ± 0.41 | 40.67 ± 0.35 |
| **DANN** (Ganin et al. (2016)) | 40.20 ± 0.50 | 56.89 ± 0.39 | 55.83 ± 0.40 | 43.25 ± 0.41 | 50.63 ± 0.38 | **58.40** ± 0.51 | 55.81 ± 0.53 | 48.90 ± 0.43 |
| **ADDA** (Tzeng et al. (2017)) | 37.40 ± 0.68 | 58.40 ± 0.73 | 60.40 ± 0.83 | 42.10 ± 0.48 | 29.40 ± 0.70 | 53.30 ± 0.49 | 45.30 ± 0.53 | 38.30 ± 0.63 |
| **c-ADDA** | 41.60 ± 0.64 | 39.65 ± 0.70 | 50.00 ± 0.52 | 46.25 ± 0.52 | 45.01 ± 0.63 | 52.14 ± 0.53 | 37.43 ± 0.60 | 43.26 ± 0.58 |
| **DTN** (Zhang et al. (2015)) | 44.13 ± 0.41 | 57.42 ± 0.42 | 55.89 ± 0.48 | 45.76 ± 0.39 | 44.53 ± 0.49 | 57.34 ± 0.35 | 52.43 ± 0.38 | 51.55 ± 0.40 |
| **c-DTN** | 45.10 ± 0.44 | 49.78 ± 0.50 | 47.43 ± 0.46 | 45.79 ± 0.48 | 49.80 ± 0.40 | 55.69 ± 0.35 | 50.10 ± 0.38 | 52.31 ± 0.29 |
| **PixelDA** (Bousmalis et al. (2017)) | **46.45** ± 0.45 | 44.33 ± 0.38 | 44.87 ± 0.41 | 46.83 ± 0.29 | 45.63 ± 0.34 | 16.37 ± 0.27 | 45.43 ± 0.35 | 47.00 ± 0.49 |
| **UNIT** (Liu et al. (2017)) | 43.88 ± 0.18 | 43.99 ± 0.23 | 44.47 ± 0.19 | 44.47 ± 0.24 | 44.47 ± 0.17 | 43.95 ± 0.21 | 44.64 ± 0.22 | 44.47 ± 0.19 |
| **DSN** (Bousmalis et al. (2016)) | 46.25 ± 0.53 | 47.50 ± 0.60 | **62.15** ± 0.58 | 39.72 ± 0.55 | 45.85 ± 0.48 | 56.65 ± 0.50 | 56.5 ± 0.38 | 42.87 ± 0.43 |
| **c-DSN** | 45.82 ± 0.53 | 44.64 ± 0.42 | 45.60 ± 0.48 | 46.32 ± 0.52 | 45.18 ± 0.47 | 45.52 ± 0.55 | 44.79 ± 0.53 | 47.37 ± 0.48 |
| **1p-DSN** | 44.12 ± 0.73 | 44.14 ± 0.20 | 45.00 ± 0.38 | 45.62 ± 0.42 | 44.78 ± 0.47 | 45.02 ± 0.65 | 44.21 ± 0.48 | 46.97 ± 0.38 |
| **mp-DSN** | 42.19 ± 0.46 | 44.70 ± 0.53 | 42.47 ± 0.48 | 40.50 ± 0.39 | 45.00 ± 0.51 | 43.80 ± 0.50 | 45.79 ± 0.48 | 42.39 ± 0.49 |
| **s-MTDA-ITA** | 44.77 ± 0.19 | 45.61 ± 0.18 | 60.00 ± 0.27 | 46.70 ± 0.28 | 49.06 ± 0.24 | 55.33 ± 0.22 | 59.90 ± 0.30 | 50.64 ± 0.26 |
| **c-MTDA-ITA** | 44.35 ± 0.27 | 42.67 ± 0.24 | 58.90 ± 0.26 | 44.32 ± 0.26 | 46.74 ± 0.22 | 54.11 ± 0.21 | 56.89 ± 0.23 | 49.64 ± 0.19 |
| **MTDA-ITA** | **46.30** ± 0.25 | **60.60** ± 0.18 | 60.50 ± 0.19 | **50.40** ± 0.20 | **55.59** ± 0.25 | 57.80 ± 0.21 | **64.20** ± 0.18 | **56.34** ± 0.20 |

**Table 6:** Classification results on Multi-PIE dataset. The best (red).

| method | P → A | P → C | P → S | A → P | A → C | A → S |
|---|---|---|---|---|---|---|
| **1-NN** | 15.28 | 18.16 | 25.60 | 22.70 | 19.75 | 22.70 |
| **ADDA** (Tzeng et al. (2017)) | 24.35 ± 2.37 | 20.12 ± 2.50 | 22.45 ± 2.11 | 32.57 ± 2.70 | 17.68 ± 2.04 | 18.90 ± 2.48 |
| **DSN** (Bousmalis et al. (2016)) | 28.42 ± 2.12 | 21.14 ± 2.08 | 2.04 ± 1.90 | 29.54 ± 1.95 | 25.89 ± 1.88 | 24.69 ± 2.08 |
| **s-MTDA-ITA** | 28.02 ± 1.59 | 21.64 ± 1.24 | 26.24 ± 1.60 | 31.06 ± 1.50 | 25.09 ± 1.40 | 25.89 ± 1.03 |
| **c-MTDA-ITA** | 25.35 ± 1.80 | 20.24 ± 1.39 | 23.64 ± 1.60 | 26.54 ± 1.33 | 20.30 ± 1.29 | 22.38 ± 1.45 |
| **MTDA-ITA** | **31.40** ± 1.55 | **23.05** ± 1.04 | **28.24** ± 1.78 | **35.74** ± 1.50 | **27.00** ± 1.25 | **28.90** ± 1.60 |

**Table 7:** Classification results on PACS dataset classification. A:Art-painting, C:Cartoon, S:Sketch, P:Photo. The best (red).

the shared space for **1p-DSN**, fig. 5f, is still evident but not as strong as in the **MTDA-ITA**, fig. 5d. This can be attributed to the lack of redundancy in the private space that helps **MTDA-ITA** to learn more disentangled shared features and usage of the target samples during training, something the **1p-DSN** fails to account for.

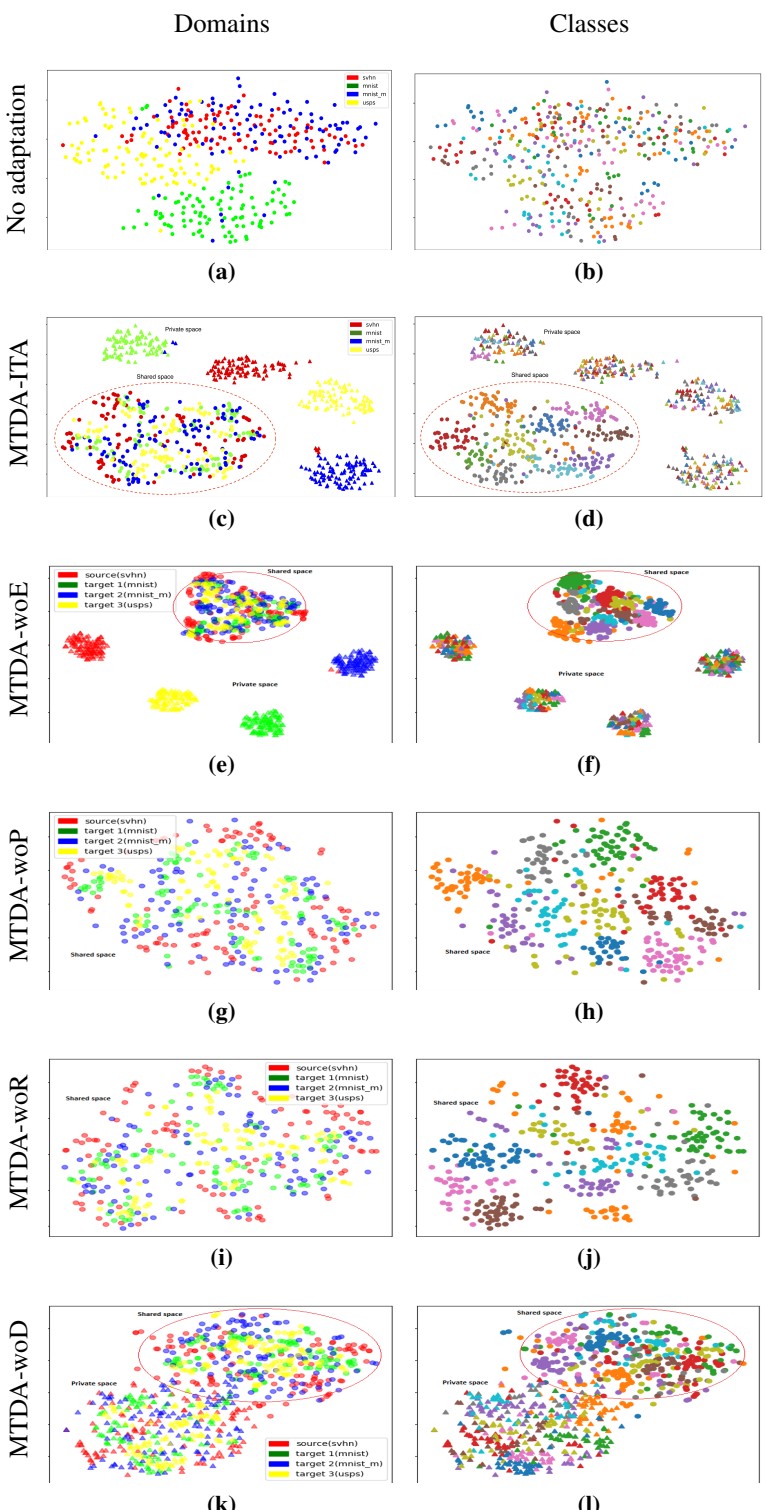

**Figure 6:** Feature visualization for embedding of digit datasets using t-SNE algorithm. The first and the second columns show the domains and classes, respectively, with color indicating domain and class membership. (a),(b) Original features. (c),(d) learned features for **MTDA-ITA** (triangle marker: private features, circle marker: shared features). Large clusters in the right column represent points from the shared space, while the smaller ones are from the private spaces. The remaining figures depict the learned features without: (e),(f) the classifier entropy loss, **MTDA-woE**; (g),(h) the private encoder, **MTDA-woP**; (i),(j) the reconstruction loss/decoder, **MTDA-woR**; and (k),(l) the multi-domain separation loss, **MTDA-woD**.