# OpenReview forum: "Unsupervised Multi-Target Domain Adaptation: An Information Theoretic Approach"
_ICLR.cc/2019/Conference_

### Official Review · AnonReviewer1 · 2018-11-02
**A Borderline Paper: the setting is interesting but the proposed approach is incremental.**

**Rating:** 5
**Confidence:** 4

**Review:**

The biggest contribution is the setting part, where one seeks to adapt one source to multiple, but somewhat similar, target domains.  It is interesting to explore such direction since in many real-world applications, applying the model to many different target domains are required.

It is also noted that there is one very related work "Multi-target Unsupervised Domain Adaptation without Exactly Shared Categories" available online (https://arxiv.org/pdf/1809.00852.pdf).  It is desirable to have a discussion and comparison with them since they are doing Multi-target Unsupervised Domain Adaptation. In their method, the exact shared category is even not required.

For the algorithm part, authors basically adopt the information-theoretic approach to handle the proposed method. This part contribution is limited since the techniques involved are very common in the domain adaptation.

---

> ### Author Response · Authors · 2018-11-26
> **Response to Reviewer 1**
>
> Thank you for the review! To improve the quality of the paper, we have made several adjustments to our paper in accordance with your review.
>
> "The contribution is limited since the techniques involved are very common in the domain adaptation".
>
> We clarify our contribution bellow:
> - We propose a novel information theoretic (IT) framework based on which a novel adversarial approach for jointly learning the shared/private features for multiple domains is proposed. Moreover, our IT framework naturally leads to utilizing the unlabeled target samples during training that is absent in most of uDA works (We did an ablation study in Sec. 4.3 on how much the information in unlabeled target samples is beneficial to the final performance). To the best of our knowledge, no other work exists that uses this combination of structures for domain adaptation, which we treat as a novel contribution.
>
> - Adaptation from one source to multiple target adaptation problem setting has been relatively under explored. To the best of our knowledge, our work is a first work addressing this DA setting, showing that jointly adapting multiple target domains in a clever way offers empirical benefit over naive solutions (combining all target datasets into single one or adapting each source-target separately). We note that Reviewer pointed to another recent work, published on arxiv 10 days prior to ICLR submission deadline, which considers a similar setting.  We contrast our approach to theirs in the comment below.
>
> - Moreover, our paper offers a new justification of why the popular auto-encoder-based regularization (derived from maximizing the mutual information between the samples and the latent features) and classifier uncertainty minimization (derived from minimizing the mutual information between the target latent features and class labels) can work for domain adaptation from an information theoretic perspective (see Sec. 2.1 for more details).
>
> - We conducted extensive experiments with detailed ablation studies on three well-known domain
> adaptation  benchmarks to validate our approach on multiple domain adaptation, demonstrating the superiority of our model over the state-of-the-art DA methods.
>
> "Desirable to have a discussion and comparison with "Multi-target Unsupervised Domain Adaptation without Exactly Shared Categories""
>
> Thanks for pointing this out.
>
> - First of all, this paper addresses the one source, multiple target domain adaptation in the context of data clustering rather than data classification. Even tough the authors of the mentioned paper use the labels of the source samples and assume that the set of all classes in target domains is a subset of source classes, it is not clear why resort to reporting clustering performance instead of the classification scores.
>
> - The paper uses the sparse representation and dictionary learning framework for domain adaptation , making it less scalable to large dataset settings compared to our approach. Indeed, this is reflected in their focus on small datasets such as Office or Yale B, which are rarely used in modern domain adaptation evaluations.
>
> - Additionally, no comparisons are reported to state-of-the-art deep domain adaptation approaches, such as ADDA, DSN, DTN, DAAN, etc.
>
> - Moreover, although this paper claims to address a new setup where the target domains not necessarily share the same categories, it assumes that the source categories contain all target categories. By this assumption, there is no difference between this setup and the standard domain adaptation setup since the classifier is trained based on the labeled source samples and the domain alignment task is independent of the target categories (due to the lack of labels in target domain).

---

### Official Review · AnonReviewer3 · 2018-11-02
**An information theoretical approach for novel multi-target domain adaptation, but not well justified.**

**Rating:** 4
**Confidence:** 4

**Review:**

This paper investigates multi-target domain adaptation which is an unexplored domain adaptation scenario compared with adapting single/multiple source to single target. A mutual information-based loss is proposed to encourage part of the features to be domain-specific while the other part to be domain-invariant. Instead of optimizing the proposed loss which is intractable, this work proposes to use neural network to model the relative functions and optimize proposed loss’ lower bound by SGD.

Method:
The proposed loss has an explanation from information theory, which is nice. However, the proposed loss is a combination of 4 different mutual information. The effectiveness of each one is unclear. An ablation study should be provided.

Clarity: The presentation should be improved, especially in the descriptions for experiments.
- Typo: Section 4: TanH should be Tanh
- Duplicated reference: Konstantinos Bousmalis, Nathan Silberman, David Dohan, Dumitru Erhan, and Dilip Krishnan. Unsupervised pixel-level domain adaptation with generative adversarial networks. In CVPR, July 2017a.

Results:
- I am confused by the experimental settings of MTDA-ITA, c-MTDA-ITA, and c-MTDA-ITA. s-MTDA-ITA. I understand c-MTDA-ITA is to combine all the target domains into a single one and train it using MTDA-ITA. And s-MTDA-ITA is to train multiple MTDA-ITA separately, where each one corresponds to a source-target pair. But I am confused by the MTDA-ITA results in both table 1 and table 2. Could the authors provide some explanation for MTDA-ITA?

- For the metric in digits adaptation, the standard metric is classification accuracy. The authors use mean classification accuracy. Is this the mean of classification accuracy of multiple runs? If so, authors should provide the standard deviation. If this is the average per-class accuracy, this is different from standard routine in ADDA, CORAL, etc.

Concerns:
The effectiveness of MDTA-ITA, s-MDTA-ITA and c-MDTA-ITA are not convincing. From the experiments, it seems the c-MDTA-ITA cannot provide convincing superior performance compared to c-ADDA and c-DTN.

---

> ### Author Response · Authors · 2018-11-26
> **Response to Reviewer 3**
>
> Thank you for the review! To improve the quality of the paper, we have made several adjustments to our paper in accordance with your review.
>
> "The proposed loss is a combination of 4 different mutual information. The effectiveness of each one is unclear. An ablation study should be provided".
>
> We  provided a detailed ablation study analyzing the effectiveness of each term in our proposed loss function in Sec. 4.3. The conclusion is that disabling each of the model's components leads to degraded performance. More precisely, the average drop by disabling the classifier entropy loss is about 3.5%. Similarly, by disabling the reconstruction loss and the multi-domain separation loss, we have about 4.5% and 22% average drop in performance, respectively. Clearly, by disabling the multi-domain separation loss, the accuracy  drops significantly due to the severe data distribution mismatch between different domains. See Sec. 4.3 for more details.
>
> "The descriptions for experiments should be improved. I am confused by the experimental settings of MTDA-ITA, c-MTDA-ITA, and s-MTDA-ITA."
>
> The experimental setups for the c-MTDA-ITA, s-MTDA-ITA and MTDA-ITA results are as follows.
>
> c-MTDA-ITA: for this case, we consider a dataset (for example SVHN) as the source and combine the others (MNIST,MNIST-M, USPS) into a single target dataset. Hence, this is a standard single source single target domain adaptation, where the target contains multiple datasets without knowing which sample belongs to which dataset (the domain label of the source samples are set to 0, and the domain label of all target samples are set to 1).
>
> s-MTDA-ITA: in this case, we consider a dataset (for example SVHN) as the source and another one (MNIST) as the target. Thus, this setup also corresponds to a standard single source single target domain adaptation, where the target contains only one dataset.
>
> MTDA-ITA: for this case, we consider a dataset (SVHN) as source and consider others (MNIST,MNIST-M, USPS) as multiple disjoint target domains. Therefore, this setup corresponds to a novel setting where we adapt jointly multiple target domains.
> It should be noted that although for both MTDA-ITA and c-MTDA-ITA, we do domain adaptation for multiple target dataset, for c-MTDA-ITA, we do not have access to the domain labels of the target datasets while for MTDA-ITA, we have access to the target domain labels (we know which target sample belong to which domain).
>
> "The meaning of mean classification accuracy".
>
> We use this term to indicate the mean of classification accuracy of five different runs using random initialization. We have included the standard deviation of the reported accuracies to the tables in the paper. Based on the standard deviation results, our model has lower variances than the other competing methods.
>
> "It seems the c-MDTA-ITA cannot provide convincing superior performance compared to c-ADDA and c-DTN".
>
> The performance scores reported in the tables indicate that c-MDTA-ITA outperforms c-ADDA (27 out of 32 cases) and c-DTN (22 out of 32 cases). More importantly, one of the contributions of our work is to demonstrate our specific, novel way of simultaneously adapting to multiple target domains offers empirical benefit over naive solutions (combining all target datasets into single domain or adapting each source-target separately in a pair-wise fashion). Our experimental results support this claim: On digit experiments, our approach ranks 1 in 9 out of 12 cases, On Multi-PIE dataset, it ranks 1 in 17 out of 20 cases. We also included  the average rank of each method over all adaptation pairs to the (last column of) tables. The scores indicate  that MDTA-ITA significantly outperforms other competing methods.

---

### Official Review · AnonReviewer2 · 2018-11-03
**limited novelty**

**Rating:** 6
**Confidence:** 5

**Review:**

In this paper, the authors proposed a new domain adaptation setting for adaptation between single source but multiple target domains. To address the setting, the authors proposed a so-called information theoretic approach to disentangle shared and private features and meanwhile to take advantage of the relationship between multiple target domains.

Pros:
- This paper conducts comprehensive empirical studies.

Cons:
- The motivation for this new domain adaptation setting is not clear to me. In the real world, the domain adaptation between multiple source domains and single target domain is in desperate need, as like human beings an agent may gradually encounter many source domains which could altogether benefit a target domain. However, I do not think that the adaptation between single source and multiple targets is intuitively in need.
- The proposed framework is quite similar to DSN, which limits this work's novelty. Though the authors take a large paragraph to illustrate the connections and differences between this work and DSN, I cannot be convinced. Especially during empirical study, the comparison is not fair. The adapted mp-DSN models multiple encoders for multiple target domains, while it is correct to extend DSN with a shared encoder for multiple target domains just like MDTA-ITA.
- There are technical flaws. The authors claimed that this work is different from ELBO optimisation, but follows an information theoretical approach. Actually, the right way to optimise the proposed loss in Eqn. (1)(2) is exactly the ELBO. Instead, the authors replace the probability/distribution q(x|z) and q(d|z) with concrete terms, which is technically wrong. such concrete term ||x-F(z;\phi)|| cannot represent a probability/distribution.
- In the experiments for feature visualisation, I do not think such comparison with original features makes any sense. The features by DSN which also separates private from shared features  should be compared.
- The presentation is in a quite poor quality, including many typos/grammatical errors.
   - The most annoying is the inappropriate citations. Every citation should be included in a brace, e.g. "the same underlying distribution Sun et al. (2016)" -> "the same underlying distribution (Sun et al. (2016))". Please kindly refer to other submissions to ICLR 2019.
   -  Typos: in the beginning of Section 2, "without loss of generalizability" -> "without loss of generality"; in the end of Page 3,  the last equation is not right, where p(y|z_s) should be q(y|z_s).
   - The font in Table 2 is too small to read.

---

> ### Author Response · Authors · 2018-11-26
> **Response to Reviewer 2**
>
> Thank you for the review! To improve the quality of the paper, we have made several adjustments to our paper in accordance with your review.
>
> "The motivation for this new domain adaptation setting is not clear".
>
> The reviewer is correct in her/his assertion that most current works focus on either pairwise or multi-source single-target domain adaptation settings.  However, the single-source multi-target setting considered here is strongly connected to many important practical problems, beyond the multi-view adaptation exemplified in this paper.  E.g., 1) Simultaneous personalization: Different target domains coincide with subjects in the test set, to which we seek adapt to.  Since all subjects exhibit e.g. similar facial expressions, it is natural to exploit data sharing across target domains. 2) Simultaneous adaptation to different scene contexts:  one seeks to adapt to  e.g. driving scenarios taken under multiple scene conditions such as illumination, seasonality, or weather.  As all target domains share common scene objects, simultaneous adaptation can leverage data sharing to improve both individual as well as global adaptation.
>
> "The proposed framework is quite similar to DSN, which limits this work's novelty".
>
> While our framework tackles an entirely different setting (single source, multi-target), it is indeed based on DSN, which itself embodies classical domain separation concepts. We go beyond DSN in several essential ways (see also discussion in Appendix C). Specifically, our loss functions, eqs 8 and 10, contain terms that encourage 1) classifier determination (low entropy, second terms) to suppress prediction of uncertain labels and 2) balanced labeling (last term) to avoid degenerate solutions where all instances in target are assigned to a single class.
>
> " Extending DSN with a shared encoder for multiple target domains just like MDTA-ITA".
>
> Thanks for pointing this out. We modified DSN to have one private encoder for multiple domains called "1p-DSN", and provided its results to the tables. As it was expected, "1p-DSN" results are quite similar to the "c-DSN", where we use one private encoder for the source domain and one private encoder for all target domains. Actually, the only difference between "1p-DSN" and "c-DSN" is that the former contains a single private encoder for all source and target domains, while the latter contains a private encoder for source domain and a private encoder for all target domains.
>
> "Authors replace the probability/distribution q(x|z) and q(d|z) with concrete terms, which is technically wrong."
>
> We used the term ||x-F(z;\phi)|| to represent the variational distributions q( x| z;\phi) as q( x| z;\phi) \propto exp(\| x - F( z;\phi)\|_1). Similarly, we model q( y| z_s) = SoftMax(C( z_s;\theta_c)),  q( d|z) = SoftMax( D( z;\psi)), where Softmax(.) denotes the softmax or normalized exponential function. we have revised the Sec. 2.1 to clarify this.
>
> "The authors claimed that this work is different from ELBO optimization, but, the right way to optimize the proposed loss in Eqn. (1)(2) is exactly the ELBO."
>
> Although both the ELBO and our work use a variational bound to make the computation tractable, by looking at the ELBO objective function H[q( z)] + E_{q( z| x)}[\ln p( x, z)] and our variational bound in Eq. 3, H( x) + E_p( x,  z)[ln q( x |  z)], show the following differences (i) in ELBO, we take the expectation of the log joint distribution p(x, z) w.r.t the variational distribution q( z| x) while in Eq.2, we take the expectation of the log variational distribution q( z| x)  w.r.t the joint distribution p( x,  z). (ii) in ELBO, we compute the entropy of the marginal distribution of the latent features q( z), while in Eq. 3, we compute the entropy of the marginal distribution of the data points p( x).
>
> "The features by DSN which also separates private from shared features  should be compared."
>
> Thanks for pointing this out. We contrasted in detail the visualization of DSN with a single private encoder to our model in Appendix H. Briefly,  both MTDA-ITA and 1p-DSN reduce the domain mismatch for the shared features and separate the shared features from private features. On the other hand, MTDA-ITA increases the domain separation for the private features while 1p-DSN is unable to enforce the private representation of different domains to be different, resulting in possible information redundancy across different private spaces.
>
> "The presentation is in poor quality, including many typos/grammatical errors. Every citation should be included in a brace. The last equation is not right, where p(y|z_s) should be q(y|z_s)"
>
> We have carefully proofread and updated the manuscript to fix typos and remove grammatical errors, as well as removed duplicate references.
>
> "The font in Table 2 is too small to read".
>
> We have moved some of results from tables 1 and 2 to the Appendix F to rectify the issue.

---

### Author Response · Authors · 2018-11-26
**Revisions to paper**

We thank the reviewers for their valuable comments. In addition to streamlining the presentation (e.g., fixing typos, improving clarity), we made the following comprehensive additions:

    1. Standard deviations of accuracies are now reported in all tables in the paper.
    2. Average rank of each method over all adaptation pairs are now reported in the tables 1, and 2 in the paper.
    3. Visualization results demonstrating the contribution of each term in our loss function to Section 4.4.
    4. Comparison (prediction accuracy) with a new variation of DSN with a single private encoder, denoted as "1p-DSN", to all the tables.
    5. Visualization experiments contrasting DSN with a single private encoder to our models in Appendix H.
    6. Discussion highlighting the relationship of our model to Information Theoretic representation learning approaches and multiple domain transfer networks in Appendices A and B, respectively.
Below we address specific comments raised by the reviewers.

---

### Meta-Review · Area_Chair1 · 2018-12-12
**Useful problem statement but incremental technical advances with modest empirical improvements**

**Confidence:** 5
**Recommendation:** Reject

**Metareview:**

The paper proposes the unique setting of adapting to multiple target domains. The idea being that their approach may leverage commonality across domains to improve adaptation while maintaining domain specific parameters where needed. This idea and general approach is interesting and worth exploring. The authors' rebuttal and paper edits significantly improved the draft and clarified some details missing from the original presentation.

There is an ablation study showing that each part of the model contributes to the overall performance. However, the approach provides only modest improvements over comparative methods which were not designed to learn from multiple target domains. In addition, comparison against the latest approaches is missing so it is likely that the performance reported here is below state-of-the-art.

Overall, given the modest experimental gains combined with incremental improvement over single source information theoretic methods, this paper is not yet ready for publication.